# Differential regulation of OCT4 targets facilitates reacquisition of pluripotency

Sudhir Thakurela [1,2,6], Camille Sindhu[1,6], Evgeny Yurkovsky[3,4,6], Christina Riemenschneider[5,6], Zachary D. Smith[1,2], Iftach Nachman[3] & Alexander Meissner [1,2,5]

Ectopic transcription factor expression enables reprogramming of somatic cells to pluripotency, albeit with generally low efficiency. Despite steady progress in the field, the exact molecular mechanisms that coordinate this remarkable transition still remain largely elusive. To better characterize the final steps of pluripotency induction, we optimized an experimental system where pluripotent stem cells are differentiated for set intervals before being reintroduced to pluripotency-supporting conditions. Using this approach, we identify a transient period of high-efficiency reprogramming where ectopic transcription factors, but not serum/LIF alone, rapidly revert cells to pluripotency with near 100% efficiency. After this period, cells reprogram with somatic-like kinetics and efficiencies. We identify a set of OCT4 bound cis-regulatory elements that are dynamically regulated during this transient phase and appear central to facilitating reprogramming. Interestingly, these regions remain hypomethylated during in vitro and in vivo differentiation, which may allow them to act as primary targets of ectopically induced factors during somatic cell reprogramming.

[1] Department of Stem Cell and Regenerative Biology, Harvard University, Cambridge, MA, USA. [2] Broad Institute of MIT and Harvard, Cambridge, MA, USA. [3] Department of Biochemistry and Molecular Biology, Tel Aviv University, Tel Aviv, Israel. [4] Raymond and Beverly Sackler School of Physics and Astronomy, Tel Aviv University, Tel Aviv, Israel. [5] Department of Genome Regulation, Max Planck Institute for Molecular Genetics, Berlin, Germany. [6]These authors contributed equally: Sudhir Thakurela, Camille Sindhu, Evgeny Yurkovsky, Christina Riemenschneider. Correspondence and requests for materials should be addressed to I.N. (email: iftachn@tauex.tau.ac.il) or to A.M. (email: meissner@molgen.mpg.de)

The direct reprogramming of somatic cells to induced pluripotent stem cells (iPSCs) has transformed the stem cell field and provided a valuable model to systematically study the molecular properties of cell state transitions[1,2]. Ectopic expression of Oct4, Sox2, Klf4, and c-Myc (OSKM) initially destabilizes the somatic program and eventually facilitates access to cis-regulatory elements linked to the activation of pluripotency-associated genes[3–7]. Current models generally divide the reprogramming process into early and late stages, with each requiring specific transcriptional co-regulators[3,8–10].

One bottleneck during reprogramming appears to be the activation of master regulators that oversee and stabilize the endogenous pluripotency network. Bulk transcriptional and epigenomic profiling suggest that the majority of responding cells remain distinct from pluripotent cells until very late in the process[7,8,11–14]. Moreover, chromatin immunoprecipitation (ChIP) of ectopically induced transcription factors (TFs) in somatic cells indicates that reprogramming factors overwhelmingly fail to bind and activate their pluripotency-related targets[15–17]. Optimized conditions or systems that facilitate near-deterministic reprogramming efficiency alleviate some of this heterogeneity[18–22]. For instance, Mbd3 depletion has been shown to result in high reprogramming efficiencies by preventing counterproductive repression of OSKM targets by the NuRD complex[13]. Alternatively, a context dependent facilitory role for Mbd3 in reprogramming has also been reported[23]. Transient priming of B-cells with ectopic C/EBPα prior to OSKM induction accelerates reprogramming by enhancing chromatin accessibility and Tet2 occupancy at regulatory regions associated with pluripotency genes[18]. Enrichment of selected subpopulations, such as SSEA-1[+]/EpCAM[+]/Sca-1[−] (Ref. [21]), is yet another approach to selectively reduce heterogeneity to facilitate molecular dissection of intermediary steps towards acquired pluripotency. Recent studies of the molecular events that occur during the exit from ground state pluripotency may also provide a new perspective to understand how this regulatory landscape becomes inaccessible to ectopic TFs when induced in somatic cells[24,25].

To complement these prior studies, we designed an experimental approach that challenges differentiating pluripotent cells to reacquire their original state under distinct conditions. Imaging-based and molecular characterization of our model system identifies a transient interval after the exit from pluripotency that permits high-efficiency reprogramming by ectopic OSKM. This high-efficiency reversion is facilitated by a set of OCT4 bound cis-regulatory elements that display a unique silencing behavior during differentiation. These regions also appear to be preferentially targeted by ectopically induced TFs in somatic reprogramming systems and retain reduced methylation levels throughout development and even into adult cells as an epigenetic remnant of the pluripotent state.

## Results

**A reprogramming barrier is set soon after pluripotency exit**. In order to specifically study the molecular barriers that separate the somatic and pluripotent state, we utilized Nanog::GFP (GFP replacing one Nanog allele and driven by the endogenous Nanog promoter) reporter containing mouse secondary induced pluripotent stem cells (iPSCs), which harbor doxycycline (dox) inducible Oct4, Sox2, Klf4, and cMyc (OSKM) transgenes. Cells were cultured in Serum/LIF, exposed to 2i/LIF media for 24h and then allowed to differentiate by switching into N2B27 media (Fig. 1a). We then collected the differentiating cells over a period of 96 h and re-seeded them as single cells in either serum/LIF (– dox) or serum/LIF with dox (+ dox) (Fig. 1a and Supplementary Fig. 1a). To quantify the efficiency of reversion to pluripotency,

we scored the number of NANOG positive colonies after an additional 96 h of growth (Fig. 1a and Supplementary Movies 1-6). As expected, removal of 2i/LIF results in rapid loss of pluripotency, leading to morphological changes and loss of NANOG signal (Fig. 1b, and Supplementary Fig. 1b-d). Cells differentiated for up to 24 h could still reacquire pluripotency with high efficiency by simply placing them in serum/LIF (– dox) condition (>70% of re-seeded cells generated NANOG[+] colonies), while those that differentiated for longer lost this potential (Fig. 1b, c). Alternatively, OSKM induction (+ dox) extends the window of high-efficiency reprogramming: after 48 h of differentiation, 86% of cells still generate NANOG[+] iPSC colonies (25th percentile: 40% and 75th percentile: 100%). Notably, differentiation beyond this window led to a similar sharp drop in reversion efficiency (Fig. 1b, 1c and Supplementary Fig. 1e-g). To ensure that our Nanog reporter allele does not affect our measurements, we repeated the experiments with a wild-type V6.5 ES cell line[26] (Supplementary Fig. 1h). To specifically define the timepoint when cells transition from high- to low-efficiency, we fitted sigmoid curves to the reversion efficiencies of both conditions at each timepoint and estimated their respective transition points (see Methods for details, Fig. 1d). This refined analysis shows that OSKM induction increases the efficient reversion time frame from ~25 h (without OSKM) to 53 h (with OSKM) of differentiation. Notably, after the second transition point, the efficiency and kinetics of iPSC colony formation resembles those observed when reprogramming from somatic cells (Fig. 1e and Supplementary Fig. 1i, j). Overall, our system reveals a transient OSKM-dependent, high-efficiency reprogramming phase during pluripotency exit that precedes irreversible commitment to the differentiated state. Importantly, we show that cells differentiated beyond the transient phase can still reprogram, but the efficiency and time required resembles those observed for mouse embryonic fibroblasts (MEFs). This suggests that a barrier similar to the one in somatic cell reprogramming is imposed shortly after the exit from the pluripotent state.

**High-efficiency reprogramming of transition state cells**. The striking resistance to OSKM induction that is acquired after 48 h may indicate a dedicated period during differentiation where cells become permanently committed to a NANOG[−] (differentiated) state. To explore how duration of OSKM induction affects the reversion behavior, we measured the fraction of NANOG[+] colonies as a function of dox duration for cells differentiated for our selected timepoints (Fig. 2a). Cells differentiated for 36–48 h required a 12–24 h pulse of dox to achieve a high-efficiency transition into transgene-independent NANOG[+] pluripotency (Fig. 2a). Live-cell imaging using the Nanog::GFP reporter showed a 24 h lag before a sharp increase in Nanog signal within the reprogramming population, which stabilized to a constant doubling rate as successfully reprogrammed lineages expanded (Fig. 2b and Supplementary Fig. 1k). Prior to the transition state, a minimal duration of dox is needed to direct cells back into the pluripotent state, which becomes subsequently consolidated by transgene-independent mechanisms.

Next, we used live-cell imaging to continuously track representative individual lineages (defined as a colony at 48 h after re-seeding) and assessed the activation of the Nanog::GFP reporter (Fig. 2c and Supplementary Fig. 1k). To account for heterogeneous or mono-allelic Nanog activation, we also performed immunofluorescence staining for NANOG after 96 h, the terminal timepoint for these experiments (Fig. 2c and Supplementary Fig. 1k). Lineages that fail to reprogram acquire a distinct morphology characterized by loosely defined patches of NANOG[−] cells that are easily distinguished from compact NANOG[+] colonies (Supplementary

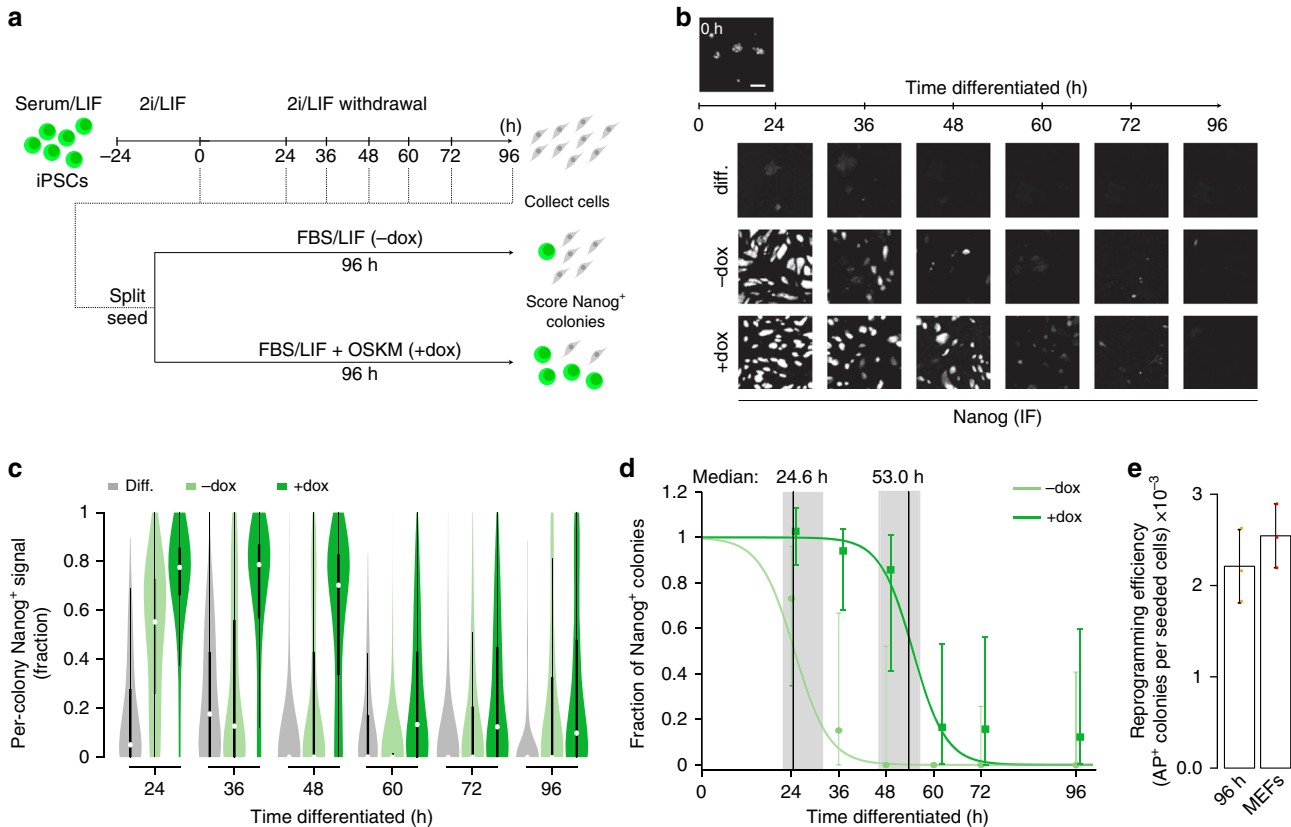

**Fig. 1** A somatic-like barrier to reprogramming is established early during differentiation. **a** Schematic of experimental system to measure reversion back to a NANOG+ state by extrinsic or intrinsic conditions. At set intervals following 2i/LIF withdrawal, single cells are re-seeded into serum/LIF, either in the absence (– dox) or presence (+dox) of doxycycline (2 μg/ml) to induce ectopic OSKM expression. The number of NANOG+ colonies was counted after 96 h. **b** Representative IF images showing the loss of NANOG signal over differentiation ("diff.", top row) and the distinct NANOG+ colony-forming efficiency under –dox and +dox conditions (imaged 96 h after the respective differentiation times; scale bar = 200 μm). All experiments were performed with n = 3 biological replicates, comprised of four technical replicates. **c** Distribution of NANOG signal (by immunostaining) within RFP-segmented colonies for differentiating cells ("diff.", gray), cells re-seeded into serum/LIF ("– dox", light green), or the serum/LIF with OSKM induction ("+ dox", dark green). Distributions are over the fraction of colony area that is NANOG positive. White dot indicates the median; boxes represent interquartile range showing central 50% of data and whiskers indicate 25th and 75th percentile data. n > 500 for 96 h and n > 9,000 for 24–72 h conditions. All experiments were performed with n = 3 biological replicates, comprised of four technical replicates. **d** Efficiency of iPSC colony-forming ability (fraction of NANOG positive pixels over the colonies) generated from IF images (panel b) and normalized to iPSC controls. NANOG+ colonies are computationally segmented, counted and normalized to the number of NANOG+ colonies generated from undifferentiated (0 h) iPSCs placed into serum/LIF. Data points represent the median values from panel **c** normalized to the control. Error bars represent the 25th–75th percentile. Grey highlighted regions indicate the 25th–75th percentile range for the estimated transition times (medians, vertical black lines) for the two conditions. Error bars for + dox are shifted to the right. Note that values are normalized ratios with respect to 0h, hence error bars may extend beyond 1. All experiments were performed with n = 3 biological replicates, comprised of four technical replicates. **e** Quantification of Alkaline Phosphatase positive colonies after 12 days of dox induction from iPSCs differentiated for 96 h or secondary MEFs generated from the same iPSC line. Reprogramming efficiency is calculated as the number of colonies per cells plated. Error bars indicate standard deviation of n = 3 reprogramming experiments in technical replicates

Fig. 1k). The frequency of successfully reprogrammed lineages over differentiation reveals a similar transition behavior between high and low-efficiency reprogramming states to the one we observed in our static population-level assays (Supplementary Fig. 1l, m). By comparing the terminal (96 h) *Nanog::GFP* reporter signal with NANOG immunofluorescence, we observed four distinct groups of colonies: NANOG+ (either GFP+ or GFP–,ab+), NANOG– (GFP–, ab–), and mixed (Fig. 2c, see Methods for details). All four colony types are substantially represented during the interval where OSKM is required to support high-efficiency reversion (Fig. 2c, d). Overall, our imaging-based lineage tracing indicates a shift from NANOG– to NANOG+ states that is most heterogeneous, in terms of reporter activation, during the transition phase (24 to 48 h) before cells go from high to low OSKM induced reprogramming efficiency (Fig. 2c, d and Supplementary Fig. 1i, j).

**Transcriptional and OCT4-binding dynamics**. To explore the molecular events that accompany this transition, we identified a total of 4616 dynamic genes over the first 96 h of differentiation by RNA-seq (Fig. 3a; Supplementary Data 1). These genes clustered into three co-regulated groups: an "early" set comprised of downregulated pluripotency-associated genes, an "intermediate" set highly enriched in chromatin regulators and a "late" set of induced genes associated with early differentiation (Fig. 3a, b and Supplementary Fig. 1n). The presence of an intermediate wave of unique regulators during 2i/LIF withdrawal may in part be linked to the distinct features of this culture condition[27] and support prior observations that cells undergo extensive epigenetic and transcriptional remodeling as they enter or exit naïve pluripotency[24,28,29]. Notably, the boundaries between early and intermediate as well as between intermediate and late cluster gene

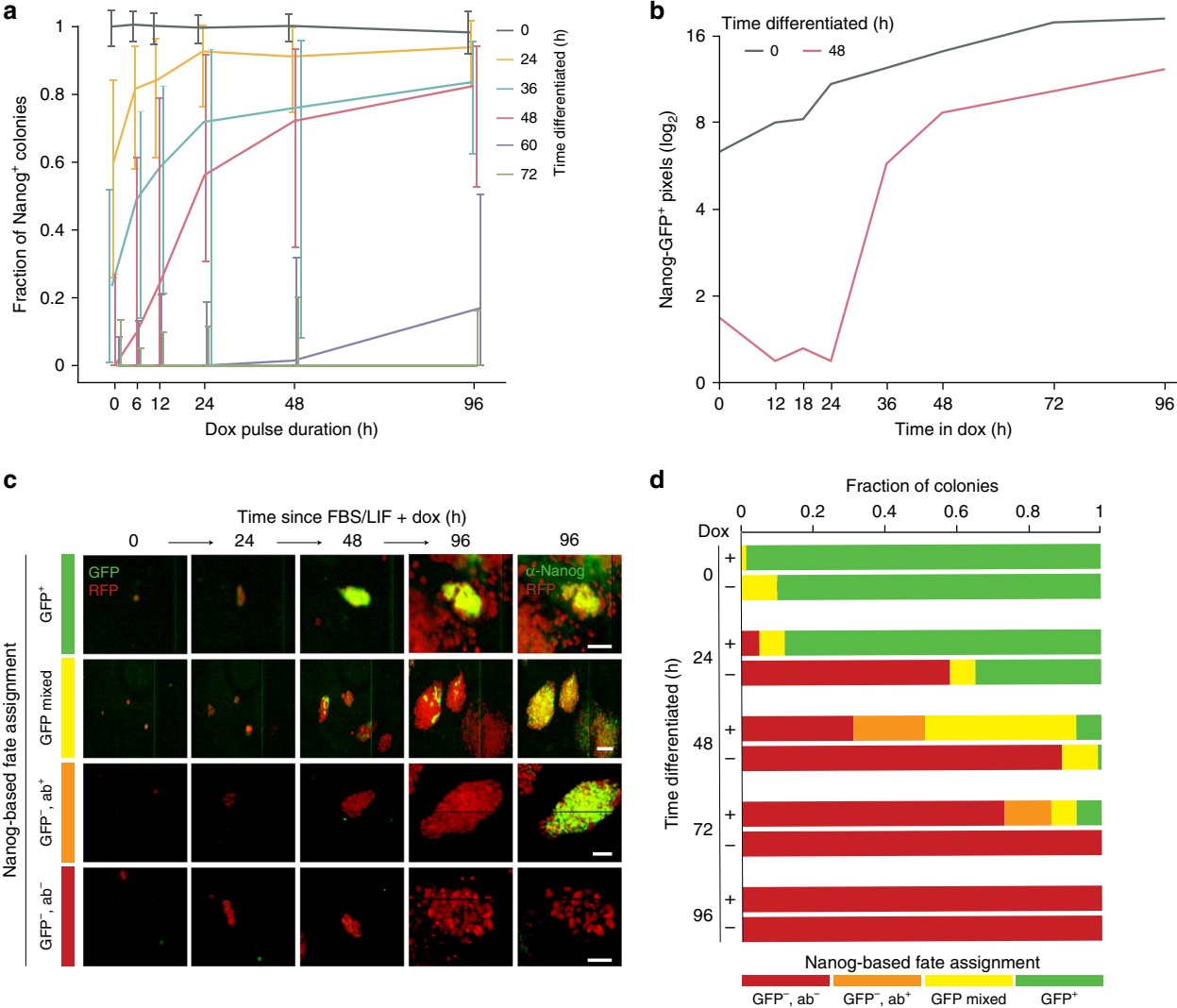

**Fig. 2** Transition from high to low-efficiency reprogramming during early differentiation. **a** Cells differentiated for our selected time intervals were seeded in serum/LIF + dox condition for the specified durations prior to dox withdrawal. The x-axis represents the dox duration and y-axis shows the normalized fraction of Nanog+ colonies after spending the rest of the 96 h experimental duration in FBS/LIF. Each colored line shows the fraction of cells for each pulse duration that were differentiated for the respective intervals. Error bars represent $25^{th}–75^{th}$ percentile. All experiments were performed with $n = 3$ biological replicates, comprised of four technical replicates. **b** *Nanog* reporter activity measured as the number of GFP$^+$ pixels (log$_2$-transformed) per imaged field for continuously tracked undifferentiated iPSCs (0h) and cells differentiated for 48 h. Undifferentiated iPSCs expand clonally, with a constant GFP signal doubling rate of 17.3 h, while the majority of cells differentiated for 48 h begin in a Nanog::GFP$^-$ state and switch to a Nanog::GFP$^+$ state between 24 h and 36 h. During this window, the GFP$^+$ area increases rapidly, with a doubling rate of 0.8 h, after which signal increases in a log-linear fashion with a doubling rate of 23.8 h. All experiments were performed with $n = 3$ biological replicates, comprised of four technical replicates. **c** For each differentiation timepoint, we performed retrospective analysis of lineages (defined as a colony formed from a re-seeded cell) from our live imaging experiments and assigned them to one of four fate outcomes: (1) Nanog::GFP$^+$ lineages activate and propagate the mono-allelic GFP reporter uniformly in all subsequent cells (green bar); (2) Nanog::GFP$^+$ mixed lineages generate heterogeneous colonies with stable propagation of Nanog::GFP signal and stain uniformly for NANOG antibody (ab) (yellow bar); (3) Nanog::GFP$^-$/ab$^+$ lineages exhibit gross cellular features of an iPSC colony without activating the endogenous *Nanog::GFP* reporter, but stain uniformly for NANOG (orange bar); (4) Nanog::GFP$^-$/ab$^-$ lineages fail to activate the reporter, do not resemble iPSC colonies, and show no signal in the NANOG antibody staining (red bar). Representative images of the four fate outcomes are shown. Scale bar = 50 μm. All experiments were performed with $n = 3$ biological replicates, comprised of four technical replicates. **d** Quantification of the four fate outcomes (same color coding as panel **c**) for −dox and +dox conditions demonstrate a sharp transition from high- to low-efficiency reprogramming responses over differentiation time that is extended for ~24 h by ectopic OSKM

expression mirror those of the OSKM-specific response remarkably well (Fig. 3c).

To further investigate the molecular drivers of the differential responses to OSKM, we performed genome-wide binding analysis of OCT4 before and during differentiation (0, 48, and 96 h after 2i/LIF withdrawal) by chromatin immunoprecipitation followed by sequencing (ChIP-seq) (Supplementary Data 2). We applied a

fold change cutoff between any two conditions to acquire a final set of 31,555 high-confidence peaks that are dynamic during this period (from 59,247 total peaks, Fig. 4a and Supplementary Data 3). OCT4 binding displayed two clear dynamics: a set of "pluripotency peaks" with specific enrichment in the undifferentiated state (0 h) ($n = 16,229$; Fig. 4a and Supplementary Fig. 2a), and a second "differentiation peak" set with minimal

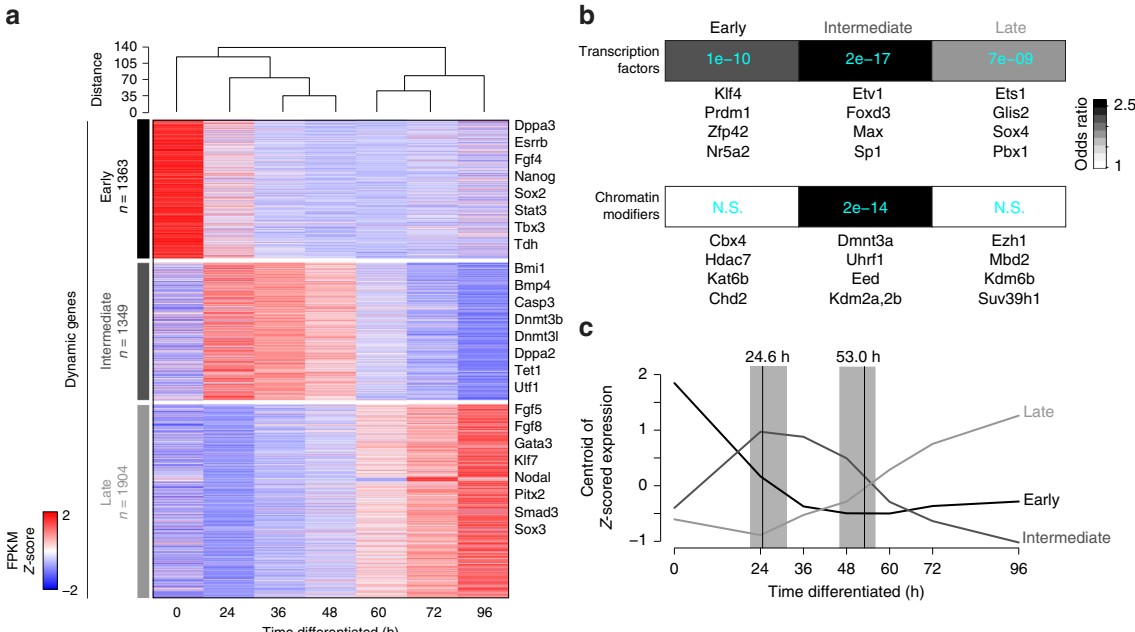

**Fig. 3** Transcriptional dynamics during the differentiation time course. **a** Heatmap of all dynamically expressed genes as measured by RNA-seq (minimum 2-fold change with FDR < 0.05 between any two timepoints; FPKM ≥ 1; n = 4,616) and clustered by k-means into three major categories: early (rapidly lost/decreasing), intermediate (transiently upregulated), or late (increasing or induced over time). Hierarchical clustering of the samples by Pearson correlation (top) separates iPSC/early differentiation timepoints and populations differentiated for longer than 48 h. Select genes belonging to each major expression dynamic category are listed on the right. Each timepoint consists of two replicates. **b** Enrichment for the three dynamic classes within a set of all annotated transcription factors (top row) or chromatin modifiers (bottom row). Greyscale represents enrichment odds ratios, while calculated P values (Fisher's exact test) are indicated in light blue text. Representative members of each set that are not already listed in panel **a** are listed underneath each tile. **c** Line plot indicating the centroid of Z scored FPKM expression values for the three separate expression dynamics (early, intermediate, and late), demonstrating the overall transcription trajectory for each group over differentiation time. For reference, the range of state transition times calculated in Fig. 1d are highlighted with grey bars, with the median of transition times indicated by vertical black lines

enrichment at 0 h but substantial occupancy in the final 96 h sample (n = 15,326; Fig. 4a and Supplementary Fig. 2a). Interestingly, OCT4 binding appears to be globally reorganized during the transient phase, which is reflected by low enrichment within either the pluripotent- or differentiation-specific sets (Fig. 4a and Supplementary Fig. 2a). The persistence of OCT4-based regulation over this timeline is in line with its continued though overall reduced expression by 96 h of differentiation (Supplementary Figs. 1c, 2b). Pluripotency and differentiation peaks localize within distinct genomic-features (Supplementary Fig. 2c, d). The pluripotency peak set appears to be associated with putative regulatory elements, as they are mostly distal to annotated transcription start site (TSSs) with CpG densities comparable to the genomic average. In contrast, the differentiation peaks primarily overlap with CpG island promoters (Supplementary Fig. 2c, d). Finally, by combining our OCT4 binding and expression data, we find that pluripotency peaks frequently occur in proximity to "early" genes that become downregulated over differentiation, while differentiation peaks are associated with "late" induced genes (Fig. 4b).

**Unique epigenetic regulation of some OCT4 target regions**. To further explore the immediate response to OSKM in our system, we began by investigating changes to expression after 48 h of OSKM induction as cells proceed through early differentiation by 2i/LIF withdrawal. At each differentiation point, we observe only subtle differences between the −dox and +dox response (Supplementary Fig. 2e). Principal component analysis (PCA) shows that the population of cells that re-establish pluripotency (0, 24, and 48 h +dox) are generally closer in expression space to the early differentiation timepoints (Supplementary Fig. 2f). We also observe a significant overlap between dox-induced genes and "early" pluripotency-genes, while dox-repressed genes are enriched for "late" differentiation-associated genes (Supplementary Fig. 2g). Although the majority of pluripotency-associated genes do not respond to OSKM induction at this timepoint (only 157 of 1363 respond; 119 up and 38 down; Supplementary Fig. 2h), those that do also lose this responsiveness shortly after 48 h of differentiation (Supplementary Fig. 2i, j).

We complemented our expression analysis with additional ChIP-seq in −dox and +dox conditions following 0, 48, and 96 h of differentiation (Supplementary Fig. 2k). Interestingly, we find a subset of 3550 pluripotency-associated OCT4 targets (pluripotency-reaccessed: henceforth referred as "reaccessed") that remain competent for ectopic binding until at least 48 h of differentiation (Fig. 4c and Supplementary Fig. 3a, b). OCT4 binding was clearly enriched in the reaccessed regions compared to the larger set of pluripotency-exclusive (henceforth referred as "exclusive", n = 9293) regions (Supplementary Fig. 3c). These two distinct subsets of pluripotency-associated peaks cannot be distinguished by their genomic distribution or overlap with notable genomic features (Supplementary Fig. 3d, e). To investigate the underlying molecular features that might result in differential OCT4 binding, we examined H3K4me2, H3K27ac, and chromatin accessibility dynamics using the assay for transposase-accessible chromatin using sequencing (ATAC-seq) over differentiation (Supplementary Figs. 2k, 3b; right panel). We observed a slight delay in the erasure of the selected histone modifications and persistent ATAC signal at reaccessed peaks compared to exclusive peaks (Fig. 4d, e and Supplementary Fig. 3b, f, g). Furthermore, during pluripotency reacquisition after 48 h of differentiation (+dox), reaccessed sites significantly gain H3K4me2, H3K27ac, and

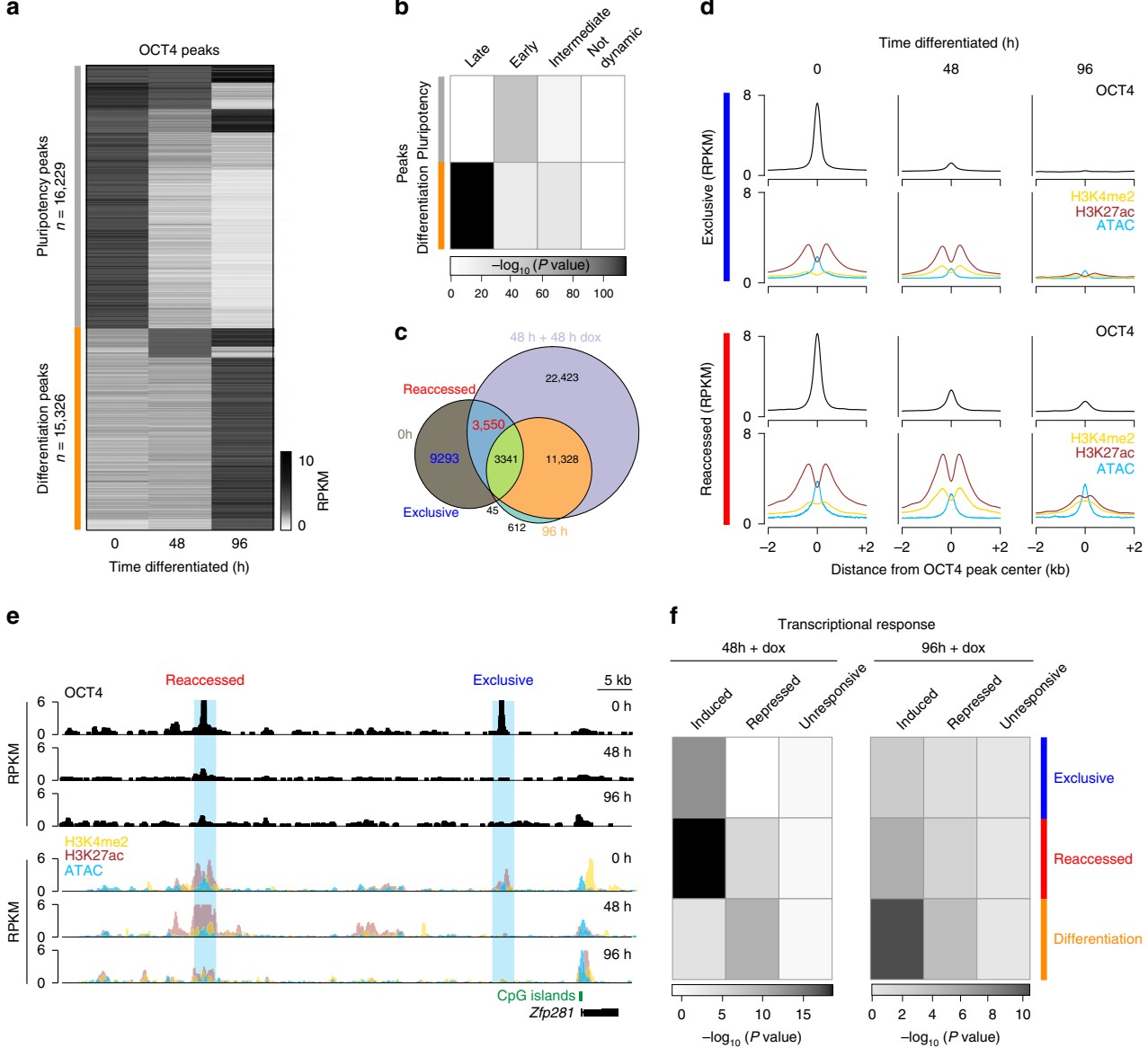

**Fig. 4** OCT4-binding dynamics distinguish several classes of gene regulatory elements. **a** Heatmap of 31,555 high-confidence OCT4-binding peaks that show dynamic changes over undirected differentiation (2-fold change between any two timepoints). Dynamic peaks were sorted based on OCT4 enrichment at 0, 48, and 96 h, separating them into pluripotency ($n = 16,229$) or differentiation ($n = 15,326$) associated peaks. RPKM: Reads per kilobase per million. Each timepoint consists of two replicates. **b** Heatmap of the statistical overlap between transcriptionally dynamic genes and genes associated with either with pluripotency or differentiation peaks. $P$ value is calculated by the hypergeometric test and displayed in $\log_{10}$. **c** Venn diagram of OCT4 peaks from iPSCs (0h; gray), the 96 h differentiated cells (orange), and cells exposed to dox for 48 h after 48 h of differentiation (light blue, 48 h + 48 h dox). Most pluripotency-associated OCT4 targets are not shared with the 96 h or 48 h + 48 h dox timepoints (pluripotency-exclusive: "exclusive", $n = 9293$ of 16,229 0 h peaks). A subset of pluripotency-associated sites (pluripotency-reaccessed: "reaccessed"; $n = 3,550$) are accessed by OCT4 in the 48 h +48 h dox sample (see also Supplementary Fig. 3a). RPKM Reads per kilobase per million. **d** Composite plots of OCT4 occupancy (black) and chromatin state at 0, 48, and 96 h differentiation timepoints for exclusive (top) and reaccessed (bottom) OCT4 targets. Chromatin state includes: H3K4me2, yellow; H3K27ac, dark red; and chromatin accessibility as measured by ATAC-seq, blue. Lines represent the median enrichment for each set within ±2 kb of the peak center. Each time-point consists of two replicates. **e** Genome browser tracks of OCT4 enrichment at the *Zfp281* locus. Top: OCT4 enrichment over differentiation (black). Bottom: selected chromatin dynamics (H3K4me2, yellow; H3K27ac, red; ATAC, blue). **f**. $P$ value heatmap of the statistical overlap between genes that are induced, repressed, or unchanged (unresponsive) upon OSKM induction after 48 or 96 h differentiation, and genes closest to each of the three dynamic peak classes. Peaks were assigned to nearest protein-coding gene, with no upper bound on distance. $P$ value for overlap significance is calculated by the hypergeometric test

ATAC-seq signal, supporting a functional response to ectopic OCT4 binding (Supplementary Fig. 3b, f, g). The set of reaccessed peaks are also more frequently proximal to genes that are transcriptionally upregulated following OSKM induction after 48 h of differentiation, and which are additionally involved in

processes related to stem cell maintenance and early development (Fig. 4f and Supplementary Fig. 3h). Overall, these data suggest that selected *cis*-regulatory elements may permit an extended opportunity for OCT4 binding, possibly because they are incompletely or differentially silenced during early differentiation.

We also sought to compare exclusive and reaccessed peak sets by their extended motif structure, which could also pinpoint unique co-regulators. By motif analysis, we find that reaccessed regions do not harbor a unique set of motifs but are generally comprised of motif combinations that are otherwise exclusive to either pluripotency or differentiation-associated peaks (Supplementary Fig. 4a, b and Supplementary Data 4). In line with their proximity to TSSs, differentiation-associated peaks contain motifs generally associated with CpG island promoters, such as from the KLF/SP1, MAZ, and EGR families (Supplementary Figs. 3e, 4a, and b). Reaccessed peaks are enriched for pluripotency-associated TFs such as SOX2 and NANOG as well as for KLF/SP1, MAZ, and EGR family motifs. Positional preference analysis for these motifs indicates that reaccessed peaks have the highest motif preference around 20 bp away from the peak center, further implicating other co-factors in stabilizing OCT4 binding during early differentiation (Supplementary Fig. 4c). As SOX2 is also essential for reprogramming and a known OCT4 co-factor, we performed SOX2 ChIP-seq during differentiation and reprogramming and compared it with OCT4 sites (Supplementary Data 5). Overall, SOX2 showed very similar binding dynamics, including its respective genomic features (Supplementary Fig. 4d-i). Comparing, OCT4 and SOX2 sites that are accessed in cells differentiated for 48 h showed a very high overlap, as around 62% of SOX2 sites are also occupied by OCT4 (Supplementary Fig. 4f). Specifically, out of 3550 OCT4 reaccessed peaks, 2496 also show high SOX2 enrichment (RPKM>2) and reducing the enrichment cutoff further (RPKM>1.5) increases the overlapping SOX2 binding set to 3112 peaks. Given this high overlap between OCT4 and SOX2 binding, we focused our subsequent analysis on OCT4.

Next, we wanted to further explore the implications of delayed OCT4 loss and persistent ATAC-seq at reaccessed sites as cells exit pluripotency in vivo. DNA methylation represents a more stable, long-term silencing mechanism with clear epigenetic properties that can influence or reflect TF binding dynamics[30]. Therefore, we investigated the methylation of CpG dinucleotides around the OCT4 target sites during early mouse development and in somatic cells, which are the typical source for reprogramming experiments. We utilized pre-existing DNA methylation data from inner cell mass (ICM, E3.5), epiblast (E6.5), ESCs and somatic tissues[31–33]. Within the ICM, both reaccessed and exclusive sets are hypomethylated, as expected for this developmental stage (Fig. 5a and Supplementary Fig. 5a). However, during the subsequent global re-methylation, reaccessed regions display a notable methylation difference in the epiblast compared to exclusive regions (0.39 and 0.69 mean methylation, respectively), and differential methylation between the two sets is preserved in ESCs and somatic tissues (Fig. 5a, b and Supplementary Fig. 5a-d). Alternatively, DNA hypersensitivity is present in ESCs but not retained in somatic cells, indicating that these cis-regulatory elements are generally not actively utilized or maintained in the canonically nucleosome-depleted, open chromatin state that is typically associated with active enhancers (Fig. 5b and Supplementary Fig. 5e). Although further investigation is needed, it is worth noting that reaccessed regions exhibit a 2.4-fold higher CpG density as compared to the genomic average, an unusual and distinguishing sequence feature (Fig. 5c). Taken together, our results suggest a route towards pluripotency that involves a small subset of pluripotency-associated genes and selected cis-regulatory elements that display unique H3K4me2, H3K27ac, and DNA accessibility dynamics during differentiation in our model, as well as a surprising epigenetic memory of the pluripotent state that persists as focal hypomethylation throughout development.

**Reaccessible regions respond early in somatic reprogramming.** Our reversion system was designed to enable a detailed molecular and temporal characterization of the somatic-like barriers that prevent high-efficiency reprogramming after pluripotency exit. As a final step, we wanted to examine how these insights may translate to somatic cell reprogramming using a number of recently published and directly relevant datasets[4,6,34–36]. In line with our somatic DNA methylation analysis (Fig. 5a, b), data from MEF reprogramming[35] indicate that OCT4 reaccessed regions begin with less methylation and are more likely to lose methylation within the early stages of OSKM induction (Fig. 5d). Using recently published ATAC-seq data[6], we categorized regions as early, middle or late based on the day they gained chromatin accessibility during MEF reprogramming, and find that the OCT4 reaccessible sites are highly enriched in the early set compared to exclusive sites (Fig. 5e). Preferential opening of the reaccessible sites is also observed during B-cell reprogramming[36] (Supplementary Fig. 5f). Finally, we also find differential OCT4-binding dynamics in MEF reprogramming[4,34] (Fig. 5f). Collectively, reanalysis of these data sets indicate that the OCT4 reaccessed regions from our model are generally responsive to OSKM induction during earlier stages of somatic cell reprogramming compared to those that are active exclusively within the pluripotent state itself.

## Discussion

Previous studies indicate that the major impediment to establishing stable pluripotency is to overcome barriers that are not surmounted until late in the reprogramming process, after which the transition to pluripotency occurs with near-deterministic kinetics[37,38]. Transitioning into this deterministic phase requires activation of the endogenous pluripotency circuitry through a yet undefined route. It is well established that inclusion of chromatin remodelers and other epigenetic modifiers in the reprogramming cocktail can alter the efficiency and kinetics of somatic cell reprogramming[13,19,20,22,39,40]. In many cases, modulating epigenetic regulators appears to act either directly or indirectly by facilitating reprogramming factor access to key pluripotency-associated loci, further implicating a regulatory or epigenetic barrier that prevents high-efficiency reprogramming from the somatic cell state. Central, long-standing questions that remain to be answered are related to the nature of this barrier as well as when and how it is overcome.

In this study, we find that a somatic-like barrier to reprogramming is set shortly after cells exit pluripotency. Establishing this barrier correlates with restricting OCT4-binding to a specific subset of pluripotent-state cis-regulatory elements. The period between when pluripotency can be restored by Serum/LIF alone and when this OSKM-resistant barrier is imposed points to a molecular determination event during early differentiation that is subsequently reflected in the low efficiency and extended latency of somatic cell reprogramming. The transient high-efficiency reprogramming response is likely accomplished through the induction of a subset of pluripotency-associated genes via a core set of distal regulatory elements that are characteristically active in the pluripotent state. OCT4's ability to access these regions correlates with the reprogramming outcome and may represent the bottleneck through which re-establishment of pluripotency is initiated. However, OCT4 engagement with this minimal subset of pluripotent-state cis-elements represents only a first step to reconfiguring pluripotency and does not induce its target genes to levels seen in stably self-renewing pluripotent stem cells. As additional components of the full pluripotency network stabilize, and the expression of key regulators increases, OCT4 may broaden its regulatory role to include the larger set of initially

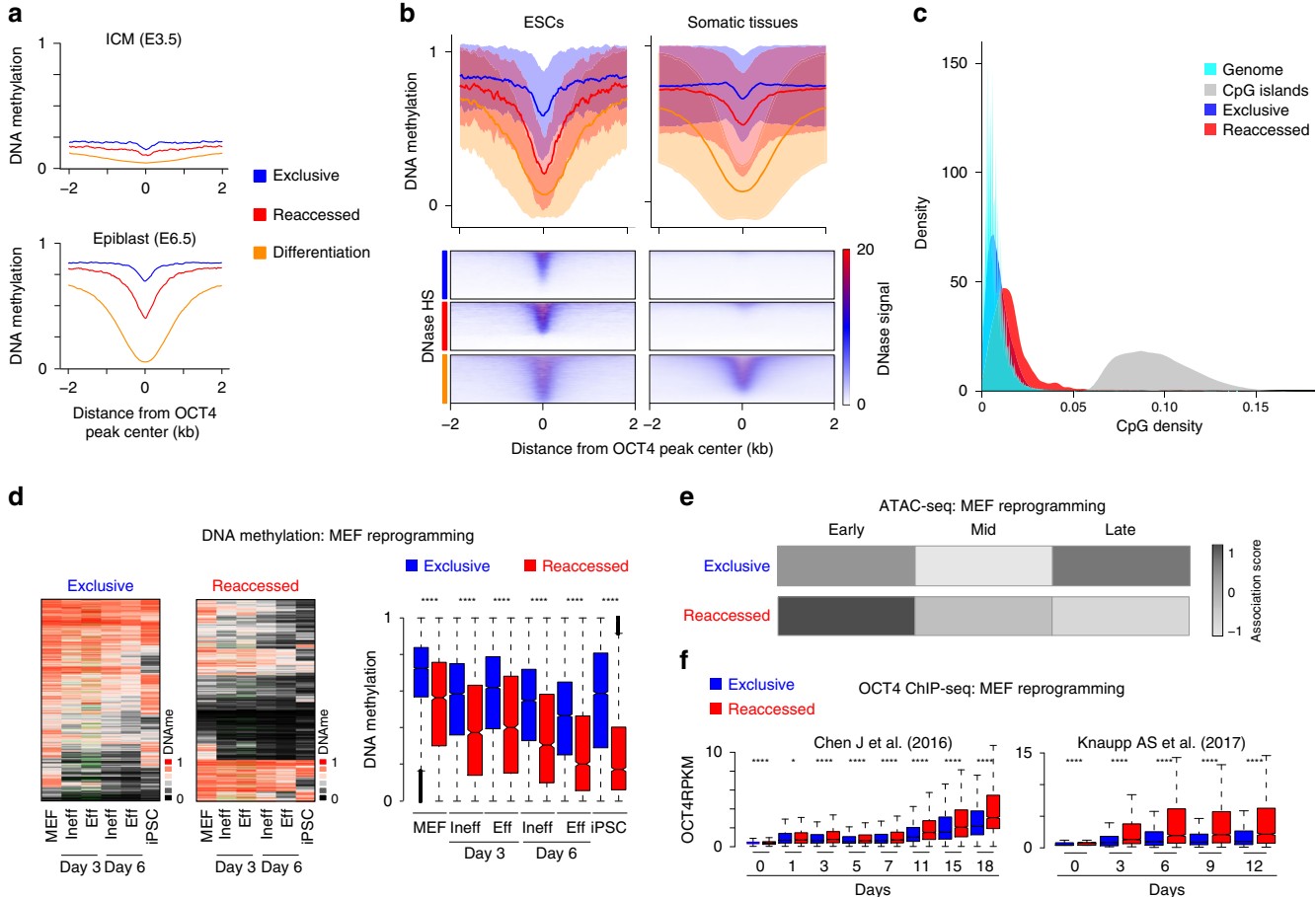

**Fig. 5** Accessible OCT4 targets retain a residual epigenetic signature over differentiation. **a** CpG methylation levels of exclusive, reaccessed, and differentiation-associated OCT4 targets during early development in vivo. Composite plots of mean DNA methylation (WGBS data) are shown for E3.5 ICM and epiblast (E6.5). Differentiation-associated OCT4 peaks are shown for comparison. **b** Reanalysis of WGBS (upper panel) and DNase hypersensitivity data from ENCODE (lower panel) for OCT4 peaks in ESCs and six somatic tissues, shown as "somatic average"[31,33,62]. Peaks were extended to ±2 kb from the center before calculating the average signal. Lines indicate the median values for each set while shaded areas represent ±1 standard deviation. **c** Histograms of CpG density within 1 kb genomic tiles, annotated CpG islands, and OCT4 peaks, either exclusive or reaccessed (mean/median CpG counts per 100 bp: 8.01/6, 67.65/54, 5.67/5, and 11.65/9, respectively). **d** Heatmaps show DNA methylation changes during MEF reprogramming at exclusive and reaccessed regions[35]. OSKM induced MEFs were divided into efficient (Eff) and inefficient (Ineff) reprogramming populations based on established markers prior to methylation profiling. On the right panel, the WGBS data is represented as boxplot. Horizontal black lines indicate medians; boxes show central 50% of data and whiskers indicate 25th and 75th percentile data. Mean methylation at our regions of interest required a minimum of 1 CpG of at least 3x coverage and green indicates insufficient coverage. *P* values are calculated using Mann-Whitney test: ****$p <$ 0.0001. **e** Heatmap of the association score (row normalized) between exclusive, and reaccessed OCT4-binding regions with regions that become accessible during early, middle or late stages of MEF reprogramming (based on a recent study[6]). The significance (the association score) is calculated using the regioneR package[60]. **f** OCT4 enrichment for exclusive and reaccessed regions during MEF reprogramming (reanalyzed ChIP-seq data from Refs [7,32]). Horizontal black lines indicate medians; boxes represent interquartile range showing central 50% of data and whiskers indicate 25th and 75th percentile data. *P* values are calculated using Mann-Whitney test: ****$p <$ 0.0001; *$p <$ 0.05. RPKM Reads per kilobase per million

inaccessible enhancer regions. It is possible that the immediately responsive elements observed in our study nucleate target gene induction to a certain threshold but are not themselves sufficient for the complete transcriptional output observed when the pluripotent network is fully activated.

Reprogramming-associated OCT4 targets are distinguishable by several notable characteristics, including delayed silencing reflected by a more persistent euchromatic signature, a higher than genomic average CpG density, and a unique combination of transcription factor binding motifs. Ectopic binding by OCT4 at reaccessed regions is distinguishable by the dual presence of co-factor motifs associated with both pluripotency and early differentiation, indicating that a combinatorial logic may dictate the differential accessibility of these regions during high- and low-

efficiency reprogramming. The transition state in our system emerges as the pluripotency network is dismantled but precedes establishment of a differentiated cellular identity. As such, it aligns principally with the recently described "formative pluripotency" stage that separates naïve pluripotency from lineage specification during gastrulation[24,41,42]. Part of the pluripotency-associated transcriptional and epigenetic program is also utilized during primordial germ cell (PGC) development[43–46] leading us to speculate that these regions could be important for PGC development. In this model, preserving germline potential within the pluripotent epiblast would require protection of critical regions during global genome remethylation, leading to a persistent epigenetic memory that is carried into rest of the soma. This residual signature of the pluripotent state appears analogous

to the proposed epigenetic memory of somatic patterning retained in iPSCs derived from different cell types[47–49]. Finally, it is worth noting that the retention of an open chromatin signature at reprogramming-associated *cis*-elements is not the sole permissive factor for deterministic reprogramming: while euchromatic signatures persist at these regions past the transition point, they are nonetheless initially refractory to OCT4 binding when ectopic OSKM is induced in cells differentiated for more than 48 h. One factor in this context may be the decreasing total OCT4 levels (endogenous plus ectopic) during the differentiation, which may affect pluripotency reacquisition along with the epigenetic state of reaccessed targets.

Our system enabled us to directly compare high-efficiency reprogramming populations with those that reprogram only at somatic-comparable rates with the goal of connecting mechanisms that allow experimentally induced changes in cell fate to those that oversee normal developmental processes. By separating exogenous (Serum/LIF) from intrinsic (ectopic OSKM) responses over differentiation time, we have effectively described moments of specification and determination in a molecularly dissectible manner. These principles may be broadly applicable as a quantitative method for describing cell state transitions.

## Methods

**Cell culture.** Doxycycline (dox)-inducible secondary induced pluripotent stem cells (iPSCs) containing GFP targeted to the endogenous *Nanog* locus[50] and constitutive nuclear RFP were maintained on CF1 mouse embryonic fibroblasts (MEFs; Applied Stem Cell, catalog # ASF-1213) in KO-DMEM containing 15% fetal bovine serum (FBS), 1% penicillin/streptomycin, 1X glutaMAX supplement (Gibco), 1% non-essential amino acid, and $5 \times 10^5$ U LIF. To perform the colony-forming assay, iPSCs were first changed into serum-free N2B27 medium supplemented with LIF and 2i (3 μM CHIR99021 and 1 μM PD0325901) for a minimum of 24 h. Prior to differentiation, iPSCs were fully dissociated to single cells and seeded at low density (~12,000 cells/cm²) on gelatin-coated plates in LIF 2i medium. After a minimum of 18 h, differentiation was induced by rinsing iPSCs thoroughly with 1X PBS and changing to N2B27 media without 2i/LIF. To test colony-forming potential, differentiated cells were trypsinized to single cells and seeded at very low density (~1300 cells/cm²) on CF1 MEFs in KO-DMEM containing 15% serum and LIF (no dox condition) or supplemented with 2 μg/ml dox (+dox condition). For the OSKM pulse assay, differentiated cells were trypsinized and seeded as described above.

**ChIP and ChIP-seq library construction.** Cells were crosslinked in 1% formaldehyde for 10 min at 37 °C with constant stirring, followed by quenching with 125 mM glycine for 5 min at room temperature. Cells were rinsed with 1X PBS and lysed, and chromatin was sheared using a Branson sonicator to a DNA fragment range of 200–1000 base pairs. Chromatin was incubated with antibody at 4 °C overnight with constant rotation. Co-immunoprecipitation of antibody-protein complexes was performed using Protein A or Protein G Dynabeads for 1 h at 4 °C. ChIP libraries were completed using previously reported method[51]. In brief, immunoprecipitated DNA was end repaired using the End-IT DNA End-Repair Kit (Epicentre), extended using Klenow fragment (3'–5' exo) (NEB), and ligated to sequencing adapter oligos (Illumina). Each library was PCR-amplified using PFU Ultra II Hotstart Master Mix (Agilent), and a size range of 300–600 base pairs selected for sequencing. Immunoprecipitation was carried out using following antibodies: OCT4 (Santa Cruz, sc-8628x; 5 ug per 10⁶ cells), SOX2 (Santa Cruz, sc-17319; 5 ug per 10⁶ cells), H3K4me2 (Millipore, 07-030; 5 ul per 10⁶ cells), H3K27ac (Diagenode, C15410196; 1 ug per 10⁶ cells). Libraries were sequenced on the Illumina Hiseq 2000.

**RNA-seq library construction.** Polyadenylated RNA was selected using Oligo dT beads (Invitrogen) and fragmented to 200–600 base pairs, then ligated to RNA adaptors using T4 RNA ligase. Libraries were sequenced on the Illumina Hiseq 2000.

**ATAC-seq library construction.** Nuclei were isolated from 50,000 cells and incubated with Tn5 transposase (Illumina) for 30 min at 37 °C. DNA was purified and PCR-amplified using customized Nextera PCR primers. A size range of 200–1000 base pairs was selected for sequencing on the Illumina Hiseq 2000.

**RNA-seq analysis.** RNA-seq reads were mapped to the mouse genome (mm10) using TopHat v2.0.14[52] with the flags: "–no-coverage-search–GTF gencode.vM4. annotation.gtf" where gencode.vM4.annotation.gtf is the Gencode vM4 reference

transcriptome available at gencodegenes.org. Cufflinks v2.2.1[53] was used to quantify gene expression and assess the significance of differential expression. Briefly, Cuffquant was used to quantify mapped reads against Gencode vM4 transcripts of at least 200bp with biotypes: protein_coding, lincRNA, antisense, processed_transript, sense_intronic, sense_overlapping. Cuffdiff was run on the resulting Cuffquant .cxb files, with a contrast file specifying comparisons between all pairs of differentiation timepoints, and all time-matched –dox/+ dox pairs. Genes were deemed to be "dynamic" if they showed a statistically significant change in expression between any two differentiation timepoints, with a minimum fold change of 2 and a minimum expression of 1 FPKM in that comparison. Expression level, fold change and statistical significance were all assessed by Cuffdiff. Genes were deemed to be "OSKM responsive" at each of four timepoints by applying the same criteria to the corresponding –dox/+dox conditions. Dynamic genes were grouped into three broad expression dynamics via k-means clustering of gene expression levels across the differentiation time course. k = 3 was chosen by varying k from 2 to 20 and looking for maximum cluster separation as assessed by the silhouette score.

**ChIP- and ATAC-seq read mapping.** Reads were mapped to the mouse genome (mm10) using Bowtie v2.2.5[54] (ref. [51]) with default options.

**OCT4 peak calling, quantification, and clustering.** OCT4-binding sites were identified using the MACS v2.1.0[55] (ref. [52]) peak caller with the flags: "–bdg–gsize mm", an FDR < 0.05 and using a common whole cell extract BAM as the background for all timepoints. Peaks were called against a set of merged whole cell extract (WCE) reads generated by randomly sampling 10M reads from six different WCE samples. In order to track the dynamics of individual peaks over time, we devised the following strategy to merge peaks from different timepoints into an epitope-wide "consensus peak set." Peak summits called by MACS were merged into a consensus region if they fell within 1 bp of each other. A new summit location was determined by taking the weighted average of all peak summits within the consensus region. Following the designation of the new summit, the peak region was defined by extending outwards by 300 bp on either side of the summit. Peak intensities were defined as the maximum number of reads within the 600 bp peak region, normalized by length and library size to get an RPKM value. Library size was taken as the number of reads mapping to non-mitochondrial chromosomes. Based on a comparison of the distribution of peak intensities from ChIP and WCE conditions, 3 RPKM was chosen as the threshold for discrete binding events. To identify relevant differential binding events, loci bound at both 0 h and 96 h, or unbound at both 0 h and 96 h, were ignored. The remaining peaks were split into three dynamics based on whether they were above threshold in the 0, 96, and 48 h +dox conditions.

**Quantification of H3K4me2, H3K27ac, and ATAC-seq.** For each OCT4 consensus peak, the Bioconductor package QuasR[56] was used to count the number of ChIP- and ATAC-seq reads within a centered 600bp window. Read counts were normalized by library size to get RPKM.

**Composite plots.** Composite plots for OCT4, H3K27ac, H3K4me2, and ATAC-seq are created using the Homer[57] package. In brief, "tag-directories" were created for all ChIP-seq and ATAC-seq samples. Peaks were extended by 2000 bp in each direction from the summit, and tag directories were then used to create a matrix for all peak sets with per nucleotide tag densities. Individual replicates were normalized for its respective sequencing depth. Matrix file containing normalized read counts within an extended 4000-bp window were imported into R to create the plots. In case of WGBS data, CpGs with less than 3× coverage were excluded. An in-house script was used to generate a matrix with mean DNAme values for regions of interest using a 100 bp window with overlapping sliding window of 25 bp. This matrix was then used for calculating ±1SD at each position and plotted using R.

**Motif analysis.** A set of 881 TF binding site motifs were obtained[58] and FIMO[59] was used to scan all OCT4 consensus peaks for occurrences of these motifs. Peaks were deemed to contain a motif if FIMO reported a p-value below 1e−4 at one or more locations within the peak. This yielded a binary matrix where rows represented OCT4 peaks, columns represented motifs, and each element represented motif presence or absence (1 or 0, respectively). Columns were grouped into clusters according to presence or absence of motifs for each peak set to identify cluster of motifs enriched in a particular peak set.

**Read-density heatmaps.** Like composite plots, generation of read-density heatmap followed similar steps as described in Homer documentation but using a window size of 10bp. The signal density was internally normalized for each sample type (OCT4, H3K27ac, H3K4me2, and ATAC) before plotting the read-density heatmaps. Minimum and maximum signal was calculated across all samples at all peak sets and this signal was then capped at 99th percentile to remove outlier that might bias color density in the heatmaps.

**Genomic and gene overlaps**. Statistical significance of gene set overlap was calculated using R package GeneOverlap (https://github.com/shenlab-sinai/geneoverlap). The significance of genomic overlaps was calculated using the regioneR[60] package.

**Image acquisition and immunohistochemistry**. All images for the colony formation, OSKM pulse, and directed differentiation assays were acquired using the Celigo S Image Cytometer. For imaging the colony formation and OSKM pulse assays, (Figs. 1b, 2a, and Supplementary Fig. 1e-g), plates were fixed in 4% paraformaldehyde and immune-stained for NANOG (BD Pharmingen, 560259) at 1:1000 dilution and detected using Alexa 488 conjugated secondary antibodies (Jackson Immunoresearch). For live imaging of lineage tracking (Figs. 1, 2c, and Supplementary Fig. 1k-m), images were acquired using a IX-71 microscope (Olympus), motorized Prior XY stage, and MetaMorph image analysis software as a 5 × 5 connected field at ×20 magnification, starting 6 h after initial seeding to allow for cells to adhere.

**Image analysis**. All images were scaled and background subtracted using the ImageJ software's "rolling ball" algorithm. Counted objects (colonies or cells) were segmented according to the constitutive RFP signal using the CellProfiler software package (Fig. 1c, d and Supplementary Fig. 1e, f)[61]. The distribution of GFP pixel intensities was calculated for all segmented objects, and thresholds for positive and negative pixels calculated separately for each assay (see below). For the colony formation assay and OSKM pulse assays, pixels were classified as GFP-positive if their intensity fell above a 20% threshold of control undifferentiated iPSC colonies. To estimate reprogramming efficiency and fate outcome on a per-colony basis from our lineage tracking images, GFP signal was measured for each imaged timepoint within the RFP-segmented area. To distinguish between GFP-positive and GFP-negative colonies, a threshold was empirically determined based on the distribution of background GFP intensities. A colony was defined as GFP-positive if its mean GFP signal was higher than the maximum value of the background distribution. The identity and number of colonies was defined at the 48 h time point. In some images acquired at later timepoints where high cell density and colony merging impeded proper segmentation, segmentation boundaries from earlier timepoints (48 h) were used for estimating colony mean GFP signal. In those instances, the segmented area was expanded to simulate colony growth. Final automatic results were corrected manually to include colonies that drifted (<10% of total colonies). To determine the onset time of GFP signal (Fig. 2b and Supplementary Fig. 1m), the GFP intensity distribution over the full imaging field was used. Colonies were classified manually into one of four possible fates outcomes based upon GFP signal (Fig. 2d).

**Modeling state transition times**. To determine the timing of transitions between the three empirically determined reprogramming response states (Fig. 1), we first estimated a distribution of possible transition times. The three biological replicates for the colony formation assay were combined and the fraction of GFP-positive pixels normalized to the undifferentiated iPSC control. The aggregate population of GFP+ fractions was split into groups corresponding to every 10th percentile separately for the + dox reprogramming condition and the + serum/LIF alone (−dox) condition. Each group was fitted separately to a sigmoid function of GFP+ fraction over differentiation time, and estimated transition times were defined as the median time where GFP+ fraction equals 0.5.

**DNase analysis**. DNase reads were downloaded from GEO (GSE49847) and mapped to the mouse genome (mm10) using Bowtie v2.2.5 with default options. DNase signal as represented in read-density heatmaps was computed as reported above.

**DNA methylation analysis**. CpG methylation tables were downloaded from GEO (GSE84236, GSE42836, GSE30206), and CpGs with coverage of less than three reads were ignored. 5 kb windows centered on OCT4 consensus peaks were subdivided into 50 bp bins, and the methylation level of all CpGs in the bin with coverage of at least 3 reads were averaged. DNA methylation data from MEF reprogramming was also processed in same way to calculate CpG methylation levels for selected peak sets.

**Reporting summary**. Further information on research design is available in the Nature Research Reporting Summary linked to this article.

## Data availability
All data generated as part of this study have been deposited in GEO under accession number GSE117205. Additional published data sets used in this study can be found under the following accession numbers: GSE84236: DNAme (ICM(3.5) and Epiblast (6.5)); GSE67520 & GSE101905 : OCT4 ChIP-seq; GSE93029: ATAC-seq MEF (reprogramming); GSE96611 : ATAC-seq (B-cell reprogramming); GSE106838 : DNAme (MEF reprogramming); GSE42836 : DNAme (Somatic tissues); GSE30206 : DNAme (mESC); GSE49847 : DNase-seq.

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

## Acknowledgements
We would like to thank all members of the Meissner and Nachman labs. In particular J. Donaghey, R. Pop, C. Galonska, A. Mohammad for experimental help and advice. M. Ziller and W. Mallard for data analysis during the early stages of the project. S. Grosswendt and J. Charlton for critical feedback on the manuscript. This work was supported by the New York Stem Cell Foundation, the NIH (P01 GM099117 and P50 HG006193) as well as the Max Planck Society.

## Author contributions
ST, CS, ZDS, IN and AM designed and conceived the study. CS, EY and CR performed experiments, ST and EY performed the data analysis. ST, ZDS, IN and AM wrote the manuscript with assistance from all authors.

## Additional information

**Competing interests:** The authors declare no competing interests.

