## [Peer Review File · Nature Communications]

Reviewers' comments:

Reviewer #1 (Remarks to the Author):

In this manuscript entitled: "Differential regulation of OCT4 targets facilitates reacquisition of pluripotency", Thakurela et al., developed an elegant model to study the OSKM-induced re-entry to pluripotency straight after iPSCs exit from pluripotency. Interestingly, the authors reveal an almost binary on/off switch to irreversibly exit pluripotency as well as to re-enter pluripotency. The authors identify a short time window after differentiation where nearly all the cell reacquire pluripotency upon OSKM induction. During this time window, a fraction of pluripotency sites can be re-accessed by OCT4. The authors therefore propose that these sites retain a "pluripotency epigenetic memory" after differentiation. Binding these pluripotency-memory sites in somatic cells by OSKM is expected to drive successful reprogramming to pluripotency.

Overall, I find the manuscript to be well written and the data clearly presented and strongly support the main conclusions of this study. I therefore recommend the publication of this manuscript in NATURE COMMUNICATIONS after addressing the following minor concerns:

- 1- It is puzzling that the OCT4 peaks in (48h + 48h Dox) overlap more with OCT4 peaks in (96h diff.) than with that of (0h), despite almost all cells become pluripotent. Are the late genes associated with OCT4 sites after 96h differentiation not responsive in (48h + 48h Dox) despite being bound by OCT4. If so, the authors need to examine whether these genes are not responsive due to culture conditions. i.e. if (48h + 48h Dox) cells grown without LIF would induce these late genes and not the pluripotency genes despite OCT4 access the re-accessed genes and therefore the (48h + 48h Dox) cells won't re-enter pluripotency.
- 2- The authors used a 1kb window to merge OCT4 peaks at different stages, this is a very loose cut off for overlap. I recommend merging OCT4 peaks which are usually 300 bp on average if they overlap by 1 bp or more and see if this will change the overall conclusion of the paper.
- 3- It was not possible to assess the quality of the sequencing data, as the authors have not provided a summary of sequencing coverage, the quality or library complexity obtained from the different samples.
- 4- The authors need to carry out motif analysis on their OCT4 ChIP-seq data as this may reveal other TFs that are uniquely associated with the re-accessed OCT4 sites.
- 5- As reprogramming efficiency is affected by OSKM protein levels, the authors need to show how similar the ectopic OSKM protein levels are once induced from the different stages of differentiation.

Reviewer #2 (Remarks to the Author):

Thakurela et al. seek to better characterise the molecular mechanism of transcription factor mediated reprogramming to pluripotency. To this end, they differentiate ES cell cultures and study the timescales with which these cultures can re-establish pluripotency with or without expression of the reprogramming OSKM (via a secondary reprogramming system). They report that efficient reprogramming is no longer possible after approximately 48hrs of differentiating (even with induction of OSKM). By way of mechanism, they study the gene expression changes (by bulk RNA-seq analysis) that occur during their differentiation procedure, but focus principally on the changes in Oct4 binding

sites by CHIP-seq analysis. They describe a subset of Oct4 binding sites which can be reaccessed by Oct4 upon re-expression of OSKM and report some distinctive features of these sites in comparison with the peaks that are found exclusively in pluripotent cells.

This initial part of the manuscript must be viewed in the context of recent studies from Austin Smith's laboratory which map the exit from 'naïve' or 'ground state' pluripotency in some detail, including the ability of exiting cells to re-establish pluripotency (Kalkan et al., 2017; Mulas et al., 2017). It is unusual that these studies are not cited in the current work, especially given the similarity in approach. Nevertheless, the use of the secondary reprogramming system during exit is novel and potentially interesting in the study of induced pluripotency – and this ambition represents the strength of the current work. However, in the manuscripts current form the exact experimental findings and methodologies are not clearly presented or described (particularly in the first half of the paper), and there appear to be methodological issues that might impact the conclusions drawn. I have a large number of concerns which are outlined below.

1. Clarity

The overall clarity of the manuscript is poor. For each dataset presented it was necessary to navigate between the text, figure, figure legend and methods to search for details of what was done. In some cases this is still not clear. It will be impossible to judge whether the data support the conclusions drawn until these issues are resolved.

This is further complicated by the use of undefined jargon. For instance, 'bi-stable system', 'switch-like behaviour' – it is not clear to me that the authors mean by this, they have certainly not experimentally validated such behaviours (if the conventional understanding of these terms), which could be attributed to heterogeneity/asynchrony of differentiation. Numerous statements which are not supported by the data should be removed – for instance 'homogenously transition', 'rapid coordinated switch' (and substantial portions of the Discussion section). The authors attempt to make very strong statements regarding mechanism, while presenting very little in the way of mechanistic data. A complete rewrite is required to enable proper assessment of the manuscript.

2. Rationale

A stated aim is to 'circumvent the heterogeneity problem'. It is not clear how their approach does this. This maybe partly in the way the manuscript is communicated (see point 1). The differentiation assay used by the authors generates a heterogenous/asynchronous cell population, as shown by their own data (for instance Figure 2D) and other labs. They show no data to support their claims regarding homogenous transitions or coordinated switches. Indeed, although the authors report a narrow time window during which reprogramming is 'deterministic' (presumably meaning highly efficient?) their own data suggests that this is an asynchronous process and they have not provided clear evidence that this approaches 100% efficiency at the single cell level.

3. Experimental system

i) Parental line. The reference cited for the parental cell line used does not give full details of the targeted allele. Does the targeted GFP reporter replace Nanog? If so, the use of a heterozygous Nanog line may well have a significant impact on the general applicability of the results observed as Nanog +/- cells differentiate more readily than wildtype (Hatano et al., 2005) and numerous studies have demonstrated an effect of Nanog dosage on reprogramming efficiency.

ii) Culture medium. The authors pre-treat FBS grown ES cell with 2i/LIF for 24hrs prior to performing the differentiation assay. This is unusual as it will almost certainly create a starting cell population

which is transcriptionally and epigenetically in flux (2i/L triggers genome-wide epigenetic changes and major transcriptomic shifts). What was the rationale for this strategy? Normally cells would be grown for at least 2-3 passages following such a major change in culture condition prior to an experimental assay.

iii) Plating density. The density of 1300 cells/cm² is an order of magnitude higher than would be routine for clonal density experiments using mouse ES cells. The very high density which results after 96hrs is evident in Figures 1b and Supplementary Figure 1h. The authors also comment on the very high density at later time points, to the extent to which it impedes segmentation. Given this high density, descriptions such as 'reseeded as single cells' and 'subsequent fate of each individual cell' seem problematic. Importantly, such high densities may well affect reprogramming leading to an underestimation of reprogramming efficiency. Could 60hr or 72hr cells reprogramming more efficiently if plated at lower density and given longer to reprogram? This is not fully addressed by Supp Figure 1g which may be impacted by cell density, but also the image does seem to show quite some accumulation of AP positive colonies after 10 days (in contrast to the authors statement in the figure legend). Figure 2a is difficult to interpret but appears to show that the fraction of Nanog + colonies is still increasing after a 96hr Dox pulse(?). This would seem to suggest that ongoing reprogramming would occur with longer treatments (contrary to the authors claims).

iv) Quantification of reprogramming efficiency. Expressing reprogramming efficiency as Fraction of Nanog positive colonies is potentially problematic, and is difficult to interpret. The absolute number of Nanog positive colonies should be shown in Figure 1c and elsewhere. The absolute number of colony numbers in each 'Nanog based fate assignment' in Figure 2d should be shown. More generally, what do the authors mean by 86% efficiency? Does this mean that they obtain around 1100 colonies? Based on their ability to track single cells can the authors provide clear data of how many single cells progress to form a Nanog positive colony following each of the differentiation time windows. How many single cells do not form colonies, and what happens to these cells? Do single founder cells often give rise to multiple colonies? This is an important point, as it impacts the interpretation of the cell state of the differentiated cells (i.e. the extent to which they represent a heterogeneous population with respect to their reprogrammability).

Comparing Figure 1d with Figure 1e it appears that the 96hrs time point reprograms with approximately 20% efficiency in 1d, but that this falls to 2% in 1e? How can this be explained? (in general, the seemingly variable systems of quantifying efficiency does not add clarity to the manuscript, and an effort should be made to more simply explain the findings).

Finally, What is the difference between 'Per-colony Nanog signal (Fraction)' (Figure 1c) and 'Fraction of Nanog+ colonies' (Figure 1d). In Supp Figure 1i, is 'Fraction seeded cells' accurate – or is this fraction of individual cells selected for analysis?

v) Single cell tracking. The statement 'continuously track representative individual lineages (defined as a colony formed from a single-seeded cell)' is unclear. How many individual cells were tracked? As single cells were plated, were all colonies not formed from single cells? If not, how often was this not the case? What does 'representative' mean? Given that the authors report the ability to track single cells, could the authors not show some representative videos or tracking files (this would help to assuage some of the concerns regarding reprogramming efficiency. Regardless of the ability to track single cells the authors should not designate this a single cell analysis, because at the density plated community effects seem highly likely).

vi) Bi-stable system. The authors do not describe what they mean by this terminology? Are they describing two different cell states that exist either side of the proposed 'transition point'? i.e. one which reprograms with 86% efficiency(48hrs) vs 20% (72hrs)? It is not clear how this nomenclature aids understanding of the molecular processes occurring at the single cell level.

vii) Bulk analysis. It is not clear how the RNA-seq analysis provides any molecular insight, and in fact is compromised by the heterogeneity issues that the authors seek to circumvent. Figure 2C and 2D indicate the high degree of heterogeneity in the cultures during the time course examined. Unfortunately, this issue applies to the Oct4 ChIP-Seq data-set also. While this is a harder problem to circumvent than the expression analysis (which could be undertaken at the single cell level or by analyzing Nanog+/- sorted populations), it should certainly impact the interpretation of this data – and the conclusions drawn should be much more circumspect than is currently the case.

Individual points and other issues:

1. Figure 1c. Why is the Nanog + signal higher in differentiated cells at the 26hr timepoint?
2. Page 3, line 98: 'kinetics of' – have the authors measured kinetics?
3. Page 3, line 104-105: 'barrier' – this is one interpretation. It is also possible that as cells differentiate they develop different requirements for reprogramming – OSKM may not be the ideal cocktail after 50hrs. Do the authors show that induction of OSKM is equivalent at each timepoint assessed?
4. Figure 2a. As above, it is not at all clear what Figure 2a depicts. It appears to show that there is a significant increase in colonies obtained following a 72hr pulse versus a 96hr pulse for all but one of the differentiation timepoints assessed. Does this not imply that reprogramming is ongoing at 96hrs?
5. Figure 2C. The lack of overlap between Nanog reporter expression and antibody staining is unusual - even in the context of the study by the Padilla-Torres laboratory. The monoallelic expression of Nanog observed in this study is now widely believed to be due to transcriptional bursting of single alleles at very low expression levels and it is unusual to see high protein expression without reporter activity. In Figure 2D as many as 20% of colonies have this pattern. It is especially unusual to see a whole colony being protein positive, reporter negative – occasional, individual cells within a colony may be more expected. Do the authors ever see GFP positive and NANOG negative cells/colonies? Further information about the construction and reliability of the reporter (as above) would be useful, as would further explanation of these unexpected observations.
6. Page 4, line 138-139. A heterogenous transition phase around this timepoint has been shown by (Kalkan et al., 2017). The slight difference in timings can easily be explained by differences in the experimental setup.
7. Oct4 ChIP. How much Oct4 protein is actually present at each timepoint assessed? The abundance of Oct4 protein after 96hrs of differentiation must be quite low?
8. In comparing the 'exclusive' versus 'reactivated' Oct4 binding sites, can the authors comment on any underlying differences in the genomic elements bound by Oct4 (promoter versus enhancer regions? Distance to TSS etc). This is quite important when considering why they have on average a higher CpG content, and also why these regions might behave differently during differentiation.
9. Page 4, Line 168. The 'global reorganization' the authors discussed is likely due to the high level of heterogeneity at this stage – Oct4 is bound to different places in the different cell types present.
10. Page 5, line 200. Is it not true by definition that Oct4 would not reaccess the pluripotency exclusive regions? The nomenclature and interpretation becomes extremely confusing here.
11. Page 5, Line 220. A clear description of the somatic tissues used is essential to the interpretation of this data – including why these make comparable and informative data sets to assess alongside the early embryonic data.
12. Page 5, Line 223-224 It appears the numerical data is inverted.
13. Page 6, Line 240 – 270. Many of the conclusions drawn here seem overstated. The data presented is correlative, and is equally consistent with the simplistic notion that sites that are accessible in the early stage of reprogramming are more likely to bind Oct4.
14. The discussion needs a substantial re-write to better represent the data presented, its caveats and correlative nature.

In particular, but not exclusively:

- Line 273 'involves' – 'correlates with' would be more appropriate
 - Line 276 'discrete molecular determination event' – what do the authors envisage this is? Why is it discrete?
 - Line 277 – 'extended latency' – have they shown this?
 - Line 279 'through induction of a core subset of pluripotency genes' – again this was a correlation. This is a hypothesis that could be tested experimentally.
 - Line 281 – 'Oct4s ability to access these regions appears to dictate reprogramming outcome' – again this is based on correlation, and in this context Oct4 is the only transcription factor for which they have assayed the binding – so it is very hard to draw such a conclusion. Again, there are range of experiments which could be undertaken to test this notion. Of note, the authors have not demonstrated that exogenous Oct4 is required for reprogramming in this system.
 - Line 295 'persistent epigenetic artefact' – this is a bizarre concept. It seems unlikely that development leaves artefacts to aid future attempts at induced pluripotency
 - Line 296-297: 'residual signature of the pluripotent state is analogous to the proposed epigenetic memory of somatic patterning retained in iPSCs derived from different cell types'. This extends the unusual suggestion above. By what criteria are these regions classified as part of a 'residual signature of the pluripotent state'.
 - Line 306-310: the conclusions drawn here are not backed up by the data presented.
15. Referencing – the literature on exit from pluripotency should be cited. In addition, the reference 24 (Page 4, Line 153) appears to be a miscitation. The referenced paper studies the transition between pluripotent states, rather than exit of pluripotency.

References:

- Hatano, S.-Y., Tada, M., Kimura, H., Yamaguchi, S., Kono, T., Nakano, T., Suemori, H., Nakatsuji, N., and Tada, T. (2005). Pluripotential competence of cells associated with Nanog activity. *Mech Dev* 122, 67–79.
- Kalkan, T., Olova, N., Roode, M., Mulas, C., Lee, H.J., Nett, I., Marks, H., Walker, R., Stunnenberg, H.G., Lilley, K.S., et al. (2017). Tracking the embryonic stem cell transition from ground state pluripotency. *Development* 144, 1221–1234.
- Mulas, C., Kalkan, T., and Smith, A. (2017). NODAL Secures Pluripotency upon Embryonic Stem Cell Progression from the Ground State. *Stem Cell Reports* 9, 77–91.

Reviewer #3 (Remarks to the Author):

In this manuscript, Sudhir Thakurela and colleagues optimized an experimental system, in which pluripotent cells were first differentiated and then reinduced into pluripotency. With this approach, the authors found a transient period before the re-establishment of pluripotency, and defined some OCT4 bounding sites with distinct epigenetic signature. This is a novel approach and will be useful for the field. However, the authors should provide some insights on how to improve our understanding to the field, thus some further work may be needed before publication.

Major concerns:

1 The authors found that, "cells differentiated beyond the proposed transient phase can still reprogram, but the efficiency and time required resembles those observed for mouse embryonic fibroblasts (MEFs)". Based on this discovery, the authors infer that ,a barrier similar to the one in somatic cell reprogramming is imposed shortly after exit from the pluripotent state. This is quite interesting. It would be great if the authors can find out what exact the barrier is and to prove the "similarity" by experiments?

2 The authors performed Oct4-ChIP-seq, and defined several sets of Oct4 targets, with a subset one termed "pluripotent reaccessed", that remain competent for ectopic binding until at least 48h of differentiation. The reviewer wants to know whether the "pluripotent reaccessed" binding sites really contribute to the barrier(s) for the somatic cell reprogramming, it's a consequence or a cause for the somatic cell reprogramming, any key gene or target was response to this process functionally?

3 The authors found the accessible peaks exhibit a 2.4-fold higher CpG density as compared to the genomic average. Can this be tested functionally to establish the relationship between them and the barriers for somatic cell reprogramming.

4 The authors checked the binding of OCT4 by Chip-seq, how about the binding pattern of Klf4 and Sox2?

5 The whole story were mainly focus on the description of the data, and discovery of some specific Oct4 accessible regions etc. Further work may be needed to validate some of those findings.

=====
Reviewers' comments:
=====

Reviewer #1 (Remarks to the Author):

In this manuscript entitled: "Differential regulation of OCT4 targets facilitates reacquisition of pluripotency", Thakurela et al., developed an elegant model to study the OSKM-induced re-entry to pluripotency straight after iPSCs exit from pluripotency. Interestingly, the authors reveal an almost binary on/off switch to irreversibly exit pluripotency as well as to re-enter pluripotency. The authors identify a short time window after differentiation where nearly all the cell reacquire pluripotency upon OSKM induction. During this time window, a fraction of pluripotency sites can be re-accessed by OCT4. The authors therefore propose that these sites retain a "pluripotency epigenetic memory" after differentiation. Binding these pluripotency-memory sites in somatic cells by OSKM is expected to drive successful reprogramming to pluripotency.

Overall, I find the manuscript to be well written and the data clearly presented and strongly support the main conclusions of this study.

We thank the reviewer for the positive feedback and suggestions.

I therefore recommend the publication of this manuscript in NATURE COMMUNICATIONS after addressing the following minor concerns:

We appreciate the constructive suggestions and we have provided new data that should address the reviewers concerns.

1- It is puzzling that the OCT4 peaks in (48h + 48h Dox) overlap more with OCT4 peaks in (96h diff.) than with that of (0h), despite almost all cells become pluripotent.

We believe the higher overlap with 96h differentiation peaks is potentially due to following reasons:

- i) During the reprogramming process we wanted to profile the transition to pluripotency and hence, unlike reprogramming efficiency calculation that were done 4 days after dox exposure, we performed the ChIP-seq experiments 2 days after dox induction. Our live imaging data suggested a 24h lag in Nanog-GFP emergence after dox induction followed by a progressive increase in signal as colonies expand. Therefore, to capture the transition state we performed ChIPs after 48h of dox. Notability, the majority of cells differentiated for 48h achieve pluripotency after 4 days of dox induction, however, after 2 days of dox they are still in transition. Therefore, 48h ChIP profiles are not necessarily expected to resemble pluripotent cells.
- ii) Furthermore, higher overlap of reprogramming cells with 96h differentiated time-point is also supported by literature as one of the essential steps during initial stages of reprogramming is to silence the differentiation related chromatin signatures.
- iii) Another important point to note is that 0h pluripotent cells are in 2i/LIF media while 48h+48h dox cells are in Serum/LIF to maintain viability (2i would not work). Previous reports have suggested binding changes when cells are grown in 2i or without the inhibitors (Galonska et al., 2015).

Are the late genes associated with OCT4 sites after 96h differentiation not responsive in (48h + 48h Dox) despite being bound by OCT4. If so, the authors need to examine whether these genes are not responsive due to culture conditions. i.e. if (48h + 48h Dox) cells grown without LIF would induce these late genes and not the pluripotency genes despite OCT4 access the re-accessed genes and therefore the (48h + 48h Dox) cells wont's re-enter pluripotency.

We indeed see that late genes associated with 96h differentiation peaks respond to OSKM induction at 48h +48dox. As shown in **Figure 2f** (left heatmap), as expected for differentiation associated genes upon OSKM induction these genes are more likely to be repressed upon OSKM

induction. On the other hand, differentiation-associated genes are more likely to be induced if cells are differentiated for 96h (**new Fig 2f, right panel**).

2- The authors used a 1kb window to merge OCT4 peaks at different stages, this is a very loose cut off for overlap. I recommend merging OCT4 peaks which are usually 300 bp on average if they overlap by 1 bp or more and see if this will change the overall conclusion of the paper.

The merging of peaks was indeed done using 1bp as suggested by the reviewer. We apologize the typographical mistake and thank the reviewer for pointing this out.

3- It was not possible to assess the quality of the sequencing data, as the authors have not provided a summary of sequencing coverage, the quality or library complexity obtained from the different samples.

We apologize that we missed to provide this information. We have now provided a new Supplementary Table (**new Supplementary table 2**) reporting the requested information. To summarize, we obtained on average 25 million total reads and 18 million average mapped reads across all samples and conditions. Overall, we overserved a very high alignment rate of around 80%.

4- The authors need to carry out motif analysis on their OCT4 ChIP-seq data as this may reveal other TFs that are uniquely associated with the re-accessed OCT4 sites.

We thank the reviewer for this suggestion and in the revised version we have included this analysis. Motif enrichment analysis for exclusive, reaccessed and differentiation associated peak sets revealed an interesting motif configuration of these regions. As expected, the exclusive peaks show enrichment for usual pluripotency associated transcription factors such as Nanog, Oct4, Sox2 while the differentiation associated peaks show enrichment for TFs that are required for different lineage diversions e.g STAT3, EGR, AP1, SP1 etc. Interestingly, the reaccessed peak set is found to be enriched in TFs from both exclusive as well as differentiation peaks (**new Supplementary Figure 4a-c**). The presence of TF motifs from both sets might be partially responsible for the ability of these regions to stay competent for OCT4 binding even after 48h of differentiation. Furthermore, a position specific motif analysis with respect to the peak center also showed a slight divergence of motif location with respect to peak center (**new Supplementary Figure 4c**). While exclusive peaks showed highest preference for the motif at the peak center, reaccessed peaks showed a shift of around 20bp for the best motif site. This flexibility in motif frequency at “reaccessed” sites might also reflect a potential role of co-factors at these regions.

5- As reprogramming efficiency is affected by OSKM protein levels, the authors need to show how similar the ectopic OSKM protein levels are once induced from the different stages of differentiation.

We thank the reviewer for the suggestion. We have now included western blots for OCT4 and SOX2, during differentiation and reprogramming (**new Supplementary Figure 1c**). This suggests proper induction for each time point and protein levels appear comparable. Unfortunately, antibodies for KLF4 and cMYC did not work well for us.

Reviewer #2 (Remarks to the Author):

Thakurela et al. seek to better characterise the molecular mechanism of transcription factor mediated reprogramming to pluripotency. To this end, they differentiate ES cell cultures and study the timescales with which these cultures can re-establish pluripotency with or without expression of the reprogramming OSKM (via a secondary reprogramming system). They report that efficient reprogramming is no longer possible after approximately 48hrs of differentiating (even with induction of OSKM). By way of mechanism, they study the gene expression changes (by bulk

RNA-seq analysis) that occur during their differentiation procedure, but focus principally on the changes in Oct4 binding sites by ChIP-seq analysis. They describe a subset of Oct4 binding sites which can be reaccessed by Oct4 upon re-expression of OSKM and report some distinctive features of these sites in comparison with the peaks that are found exclusively in pluripotent cells.

This initial part of the manuscript must be viewed in the context of recent studies from Austin Smith's laboratory which map the exit from 'naïve' or 'ground state' pluripotency in some detail, including the ability of exiting cells to re-establish pluripotency (Kalkan et al., 2017; Mulas et al., 2017). It is unusual that these studies are not cited in the current work, especially given the similarity in approach. Nevertheless, the use of the secondary reprogramming system during exit is novel and potentially interesting in the study of induced pluripotency – and this ambition represents the strength of the current work. However, in the manuscripts current form the exact experimental findings and methodologies are not clearly presented or described (particularly in the first half of the paper), and there appear to be methodological issues that might impact the conclusions drawn. I have a large number of concerns which are outlined below.

We thank the reviewer for the feedback and provide a detailed response below. We apologize that we missed to cite those papers. We have now cited and discussed them on pages 2 and 3. We hope that these responses will answer the questions and will be satisfactory to the reviewer.

1. Clarity

The overall clarity of the manuscript is poor. For each dataset presented it was necessary to navigate between the text, figure, figure legend and methods to search for details of what was done. In some cases this is still not clear. It will be impossible to judge whether the data support the conclusions drawn until these issues are resolved.

We understand that at times due to space restrictions we cannot provide all the relevant information in main manuscript text and hence we have provided some of the detailed information in either methods section or figure legends. Considering range of opinions regarding clarity of the manuscript and after carefully rereading it, we believe that overall the manuscript is understandable. But we will work with the editor on the final presentation and ensure maximal clarity. Wherever possible we have attempted to improve the writing and to provide any relevant information that was previously missed.

This is further complicated by the use of undefined jargon. For instance, 'bi-stable system', 'switch-like behaviour' – it is not clear to me that the authors mean by this, they have certainly not experimentally validated such behaviours (if the conventional understanding of these terms), which could be attributed to heterogeneity/asynchrony of differentiation. Numerous statements which are not supported by the data should be removed – for instance 'homogenously transition', 'rapid coordinated switch' (and substantial portions of the Discussion section). The authors attempt to make very strong statements regarding mechanism, while presenting very little in the way of mechanistic data. A complete rewrite is required to enable proper assessment of the manuscript.

We apologize if we misused some of the terminology, which was our attempt to help describe the cell behavior to the reader. We do agree that in strict sense our system does not confer to a bi-stable system, therefore, we have modified the manuscript text to reflect our observations more accurately. In our system during the early (< 24h) and the late (> 48h) stages of differentiation the reversion fate choices do not vary with OSKM induction as they exhibit all or none behavior respectively (**Figure 2a**). During the transient phase, the fraction of NANOG⁺ colonies and Nanog-GFP signal increases non-linearly with respect to the duration of OSKM pulse and hence depending on the time and amount of input signal cells can either differentiate or revert to pluripotency. These experiments provide evidence that cells in the transient state respond to OSKM signal as expected from an on/off (switch-like) system. We are open to suggestions from the reviewer but now use switch-like behavior similar to other reprogramming studies (Liu et al., 2016; Zviran et al., 2019) that described cell transitions to pluripotency or failure to acquire pluripotency *en-masse*. We observe a similar phenomenon as cells before the transition can

reprogram with very high efficiency while after that they behave like somatic cells and majority of cells fail to reprogram.

As mentioned above, we believe that we have provided sufficient details that support the usage of terminologies like “homogenous transition” and “coordinated switch” in **Figure 1, 2 and Supplementary Fig 1**. In brief, homogenous transition is supported by the observation that almost 90% of transient cells result in pluripotent colonies and as cells either revert or not in response to OSKM induction supports a coordinated switch. We have further explained the terms in manuscript text.

2. Rationale

A stated aim is to ‘circumvent the heterogeneity problem’. It is not clear how their approach does this. This maybe partly in the way the manuscript is communicated (see point 1). The differentiation assay used by the authors generates a heterogenous/asynchronous cell population, as shown by their own data (for instance Figure 2D) and other labs. They show no data to support their claims regarding homogenous transitions or coordinated switches. Indeed, although the authors report a narrow time window during which reprogramming is ‘deterministic’ (presumably meaning highly efficient?) their own data suggests that this is an asynchronous process and they have not provided clear evidence that this approaches 100% efficiency at the single cell level.

We agree that our differentiation might generate a heterogenous population. We are not making any claims regarding homogeneity of the differentiating cells as in a differentiating population all cells may not be tightly synchronized. However, in our assays the focus is on the outcome of the reprogramming where we observe homogenous transitions to pluripotency. We agree that reprogramming efficiency is not 100%, however, we have provided several experimental data showing that transition state cells show a very high reprogramming efficiency with around 90% of cells giving rise to pluripotent colonies.

We don’t refute the potential that these cells are heterogeneous, they are merely homogeneous in terms of their ability to generate Nanog⁺ or Nanog⁻ cells in response to exogenous conditions. Therefore, homogenous transition is with reference to the reprogramming efficiency of transient cells, which, as stated above, is around 90%. **Figure 2d** indeed shows a heterogenous response as far as the expression of Nanog is concerned, however ultimately most cells give rise to pluripotent colonies (48h; combined orange, green and yellow from **Figure 2d**) colonies after 96h. Some of this heterogeneity (manifested as colony-to-colony differences) may be due to heterogenous differentiation response of the iPSCs, while some (manifested both as between-colony differences and as intra-colony variability) as heterogenous response over time to dox induction.

As for temporal asynchrony, the dynamics in **Fig. 1** could be explained by asynchronous transition of cells from state A (responsive) to state B (non-responsive). However, the gene expression patterns argue against this possibility and support the notion that the transient phase (between ~24h and ~53h) is an actual distinct phase, with its own unique expression pattern (that cannot be explained by some temporal averaging of the early and late phases). For example, the “Intermediate” cluster (**Figure 3a**) genes are high in the intermediate phase, but low in both the early and late phases. Therefore, we believe that our observations from imaging and live-cell tracking experiments do provide enough evidence suggesting a homogenous transition.

3. Experimental system

i) Parental line. The reference cited for the parental cell line used does not give full details of the targeted allele. Does the targeted GFP reporter replace Nanog? If so, the use of a heterozygous Nanog line may well have a significant impact on the general applicability of the results observed as Nanog +/- cells differentiate more readily than wildtype (Hatano et al., 2005) and numerous studies have demonstrated an effect of Nanog dosage on reprogramming efficiency.

We apologize for not clarifying the genetic status of the cell line. One *Nanog* allele is indeed replaced by a GFP reporter allele and hence, the cell-line is heterozygous. This cell line has been used to assess reprogramming efficiency and further information can be obtained from (Buganim et al., 2012; Wernig et al., 2007). However, to further validate our results we replicated our experiments using a *Nanog* wild-type (homozygous) cell line in similar conditions as used for our secondary iPS cell line. As seen from new **Supplementary Figure 1h** we can show that heterozygous *Nanog* doesn't appear to influence the reprogramming process in our set-up and we observe a similar transient state showing very high reprogramming efficiency.

ii) Culture medium. The authors pre-treat FBS grown ES cell with 2i/LIF for 24hrs prior to performing the differentiation assay. This is unusual as it will almost certainly create a starting cell population which is transcriptionally and epigenetically in flux (2i/L triggers genome-wide epigenetic changes and major transcriptomic shifts). What was the rationale for this strategy? Normally cells would be grown for at least 2-3 passages following such a major change in culture condition prior to an experimental assay.

Our rationale was to make the starting population more homogenous through the use of 2i/LIF. However, it is not possible to return the reverting cells into 2i, as the PD inhibitor will add selective pressure on the differentiated cells (Silva et al., 2008). The transient exposure to 2i/LIF is sufficient in our experience (Galonska et al., 2015).

iii) Plating density. The density of 1300 cells/cm² is an order of magnitude higher than would be routine for clonal density experiments using mouse ES cells. The very high density which results after 96hrs is evident in Figures 1b and Supplementary Figure 1h. The authors also comment on the very high density at later time points, to the extent to which it impedes segmentation. Given this high density, descriptions such as 'reseeded as single cells' and 'subsequent fate of each individual cell' seem problematic. Importantly, such high densities may well affect reprogramming leading to an underestimation of reprogramming efficiency. Could 60hr or 72hr cells reprogram more efficiently if plated at lower density and given longer to reprogram? This is not fully addressed by Supp Figure 1g which may be impacted by cell density, but also the image does seem to show quite some accumulation of AP positive colonies after 10 days (in contrast to the authors statement in the figure legend).

The reviewer is correct that plating densities are important. Our density was carefully selected to allow reprogramming but also provide enough statistical power for our analysis in **Figure 1** and **Figure 2**. At later time-points we use appropriate measures to overcome the segmentation problem which we report in our manuscript on page 12. For 96h where colony segmentation is a potential issue, we actually define the colonies at 48h and utilize that information at later time-point. This is further explained in the methods section. Irrespective of that, we can track the individual cells to the outcome efficiently and hence, usage of the terms seems justified.

Furthermore, to explore the effect of different densities on reprogramming efficiency, we repeated our pluripotency reacquisition assay using different seeding densities. We used 88, 500, 1300 and 2600 cells/cm², which covers a range of densities lower and higher than our original one. After re-seeding, reversion efficiency was calculated with or without OSKM induction by AP-staining. As shown in new **Supplementary Figure 1a**, we did not observe any major differences in reprogramming efficiency. As evident from the data, cells differentiated for 60h or more have almost no difference in their reprogramming efficiency when seeded at different densities. Therefore, the cell density (and range around it) appears to be not affecting the reprogramming efficiency.

Lastly, we would like to point that the more direct comparison is **Supplementary Figure 1i**, where the density of AP+ colonies is similar between MEFs and 96h cells. In **Supplementary Figure 1j** we see an increase in AP signal, but this increase is due to colony growth and not additional colonies. Therefore, the apparent accumulation that is seen at 96h after 10 days of dox is not because of additional colonies but is a result of expansion of colonies.

Figure 2a is difficult to interpret but appears to show that the fraction of Nanog + colonies is still increasing after a 96hr Dox pulse(?). This would seem to suggest that ongoing reprogramming would occur with longer treatments (contrary to the authors claims).

We have now improved the explanation of the plot in text as well in the figure legend. The x-axis represents the time duration for which cells are exposed to dox and on y-axis shows what is the normalized fraction of Nanog+ colonies with respect to time point 0. Each colored line shows the fraction of cells, for each pulse duration, that were differentiated for the mentioned intervals i.e. 0, 24, 36, 48, 60 and 72 hours. We would like to clarify that the gain of Nanog+ colonies with dox for 96h is mainly observed for the transient phase i.e. cells differentiated for 36h or 48h. The gain is very minimal for time points after the transient phase (i.e. cells differentiated for 60h and 72h) and overall the observed reprogramming efficiency is very low compared to the transient phase where up to 90% cell give rise to NANOG+ colonies. Therefore, we agree that reprogramming is still occurring after 60h, however, along with our other observations in **Supplementary Figure 1i** and **1j** increase in dox exposure does not lead to higher efficiency. Hence, we believe our claims are in line with the observations.

iv) Quantification of reprogramming efficiency. Expressing reprogramming efficiency as Fraction of Nanog positive colonies is potentially problematic, and is difficult to interpret. The absolute number of Nanog positive colonies should be shown in Figure 1c and elsewhere. The absolute number of colony numbers in each 'Nanog based fate assignment' in Figure 2d should be shown.

For the analysis of any data that may involve variations based on cell counts and different replicates we used normalized counts, so that different raw/absolute numbers are comparable. Furthermore, signal from imaging may vary based on the size of the colony and hence it is required that we normalize the signal from different colonies. We devised a method to normalize based on the background intensity which is fully explained in the methods section. Therefore, we do not believe that such analysis is potentially problematic. However, to be completely transparent with respect to our experiments and observations we have provided these absolute numbers now in the manuscript in the figure legends. The absolute numbers for **2d** (totals for each time point) are already provided in **Supplementary Fig. 1k**.

More generally, what do the authors mean by 86% efficiency? Does this mean that they obtain around 1100 colonies?

By 86% reprogramming efficiency we mean that 86% of cells seeded after differentiation gave rise to Nanog+ colonies, as randomly sampled in the imaged field of views. A colony was denoted as Nanog+ as described in the methods sections and for convenience we paste that information here also. All images were scaled, and background subtracted using the ImageJ software's "rolling ball" algorithm. Counted objects (colonies or cells) were segmented according to the constitutive RFP signal using the CellProfiler software package. The distribution of GFP pixel intensities was calculated for all segmented objects, and thresholds for positive and negative pixels calculated separately for each assay. For the colony formation assay and OSKM pulse assays, pixels were classified as GFP-positive if their intensity fell above a 20% threshold of control undifferentiated iPSC colonies. To estimate reprogramming efficiency and fate outcome on a per-cell basis from our lineage tracking images, GFP signal was measured for each imaged time point within the RFP-segmented area. To distinguish between GFP-positive and GFP-negative colonies, a threshold was empirically determined based on the distribution of background GFP intensities. A colony was defined as GFP-positive if its mean GFP signal was higher than the maximum value of the background distribution. In some images acquired at later time points where high cell density and colony merging impeded proper segmentation, segmentation boundaries from earlier time points were used for estimating colony mean GFP signal. In those instances, the segmented area was expanded to simulate colony growth. Final automatic results were corrected manually to include colonies that drifted (<10% of total colonies). To determine the onset time of GFP signal, the GFP intensity distribution over the full imaging field was used. Colonies were classified manually into one of four possible fates outcomes based upon GFP signal.

Based on their ability to track single cells can the authors provide clear data of how many single cells progress to form a Nanog positive colony following each of the differentiation time windows. How many single cells do not form colonies, and what happens to these cells? Do single founder cells often give rise to multiple colonies? This is an important point, as it impacts the interpretation of the cell state of the differentiated cells (i.e. the extent to which they represent a heterogeneous population with respect to their reprogrammability).

We now provide additional live imaging videos, tracking individual cell lineages under dox or no-dox conditions following differentiation (24, 48 and 96 hours). We have manually analyzed several randomly selected representative fields to accumulate sufficient statistics on the fate distribution of individual cell lineages that have successfully landed following the re-seeding process. As can be seen in the movies, splitting of single lineage to several colonies is not common. We do get single cells merging into single colonies occasionally, mostly in the earlier differentiation points. For this reason, we define the identity and number of colonies (reported in **Supplementary Fig. 1k**) at the 48hr time point, and the fractions reported in **Figure 2d** are with respect to this total. In this analysis, we account for colonies that have merged between 48-96 hrs, meaning we still count them as separate colonies, using the time series information, as explained above and in the Methods. For example, in the 48 +dox condition, we have segmented 71 colonies at the 48hr time point of the reversion time series, based on the RFP signal. The fractions shown in **Figure 2d** (in this case around 25% GFP⁻,ab⁻, 17% GFP⁻,ab⁺, etc) are out of these 71 colonies.

Comparing Figure 1d with Figure 1e it appears that the 96hrs time point reprograms with approximately 20% efficiency in 1d, but that this falls to 2% in 1e? How can this be explained? (in general, the seemingly variable systems of quantifying efficiency does not add clarity to the manuscript, and an effort should be made to more simply explain the findings).

Figure 1d and **1e** report two different experiments and their respective measurements are computed differently. **Figure 1d** shows fraction of Nanog⁺ colonies (determined by Nanog immunostaining) normalized to the number of iPSC controls (i.e. 0h differentiation) seeded in the same density and conditions. **Figure 1e** is computed as AP⁺ colonies per cell plated. We now provide a better explanation of this in the figure legend (page 14). The difference that is observed (and in particular the high value of ~20%) is due to the cutoff we used to define a colony as reprogrammed in two totally different experiments, as well as the normalization applied in **Figure 1d** as opposed to **1e**, hence the gap in numbers. We believe it is helpful to report different measurements showing similar results to substantiate the observations. Furthermore, different experiments also require distinct ways of representations to make sure that data is presented properly. One of the reasons to add **panel 1e** was to show the ~20% quantified in the normalized-by-comparison-to-iPSC (**panel 1d**, which we believe is a more accurate way to quantify the process) “translates” to the “standard” 2% commonly observed as reprogramming efficiency in similar assays.

Finally, What is the difference between ‘Per-colony Nanog signal (Fraction)’ (Figure 1c) and ‘Fraction of Nanog⁺ colonies’ (Figure 1d). In Supp Figure 1i, is ‘Fraction seeded cells’ accurate – or is this fraction of individual cells selected for analysis?

Per-colony signal is the percent of positive pixels in each colony. When we apply a threshold to this percent, we define a colony as positive or negative. This is what was used in **Figure 1d**. The **Supplementary Figure 1i** is indeed showing “fraction of seeded cells”. We have corrected the legend to reflect the change.

v) Single cell tracking. The statement ‘continuously track representative individual lineages (defined as a colony formed from a single-seeded cell)’ is unclear. How many individual cells were tracked? As single cells were plated, were all colonies not formed from single cells? If not, how often was this not the case? What does ‘representative’ mean? Given that the authors report the ability to track single cells, could the authors not show some representative videos or tracking files

(this would help to assuage some of the concerns regarding reprogramming efficiency. Regardless of the ability to track single cells the authors should not designate this a single cell analysis, because at the density plated community effects seem highly likely.

By that statement we refer to the fact that this is a prospective analysis of colonies (lineages) formed from individually seeded cells. Colonies can be termed as “lineages” as they start from single seeded cells and finally give rise to either Nanog+ or Nanog- colonies. The number of individual cells tracked at each time point is shown in **Supplementary Figure 1k**. Almost all colonies were formed from single cells, however, it is possible that very rarely multiple cells give rise to a colony. This phenomenon is expected to be rare and the random possibility of multiple cells giving rise to a colony is expected to be same for all time points. Hence, such behavior is unlikely to affect the overall observations. Representative filmstrips are shown of selected lineages tracked by live imaging. As suggested by the reviewer, we now provide some videos showing tracking of cells during reprogramming. We do agree that we cannot call it as “single cell analysis”, which we are not as we only say that we can track individual cells and define the outcome. Furthermore, as shown above by new experiments that using different cell densities the reprogramming efficiency is not affected, we believe we can term our analysis as tracking of individual lineages.

vi) Bi-stable system. The authors do not describe what they mean by this terminology? Are they describing two different cell states that exist either side of the proposed ‘transition point’? i.e. one which reprograms with 86% efficiency(48hrs) vs 20% (72hrs)? It is not clear how this nomenclature aids understanding of the molecular processes occurring at the single cell level.

As explained above, we have now edited the manuscript and do not describe our system as “bi-stable” system. Our aim was to describe a transient state of differentiating cells, where cells can initially reprogram with high efficiency and once they passed this phase the reprogramming efficiency drops to somatic levels. During the reprogramming we seed at low density so that we can track individual cells and their fates. Seeding in low density provides us a way to track each cell and helps delineation of potential outcomes for individual cells (successful or unsuccessful reprogramming).

vii) Bulk analysis. It is not clear how the RNA-seq analysis provides any molecular insight, and in fact is compromised by the heterogeneity issues that the authors seek to circumvent. Figure 2C and 2D indicate the high degree of heterogeneity in the cultures during the time course examined. Unfortunately, this issue applies to the Oct4 ChIP-Seq data-set also. While this is a harder problem to circumvent than the expression analysis (which could be undertaken at the single cell level or by analyzing Nanog+/- sorted populations), it should certainly impact the interpretation of this data – and the conclusions drawn should be much more circumspect than is currently the case.

Bulk analysis of RNA-seq data compliments the observations made using imaging and tracking experiments. As observed by these experiments, bulk RNA-seq of the time-course also shows that the cells in the transient state have very distinct transcriptome. Even if the differentiation is heterogeneous, we still see a clear expression pattern during the differentiation time course (**Fig. 3a**), with distinct genes upregulated in the intermediate phase. The heterogeneity referred to by the reviewer is related to dox response and Nanog expression (**Fig. 2d and Supplementary Figure 1k**) is substantial only at the 48hr +dox condition.

Individual points and other issues:

1. Figure 1c. Why is the Nanog + signal higher in differentiated cells at the 26hr timepoint?

The difference between differentiation and -dox state at 36h is minimal which may arise due to technical variations. This difference is not significant.

2. Page 3, line 98: ‘kinetics of’ – have the authors measured kinetics?

We have used the term kinetics in a generalized sense referring to the change in state which have been commonly used in several previous studies including in (Kalkan et al., 2017) as they refer to measurement of changes happening in expression of proteins over time. In our manuscript the term kinetics is mainly used when we compare the rate of formation of iPSC colonies in our system with respect to somatic cells. We have shown how cells from one state can transition into other based on input duration hence, we believe that usage of term kinetics makes sense here but can adjust this if the reviewer still disagrees.

3. Page 3, line 104-105: 'barrier' – this is one interpretation. It is also possible that as cells differentiate they develop different requirements for reprogramming – OSKM may not be the ideal cocktail after 50hrs. Do the authors show that induction of OSKM is equivalent at each timepoint assessed?

We are also trying to make the point that OSKM cannot reprogramming cells efficiently once they are differentiated for a longer time duration. Cells during differentiation undergo different commitment stages. Before the transient point cells have undergone a specification event from where they can revert when subjected to right conditions, however, once the transient phase is passed cells have undergone a determination event and are committed to differentiate. Therefore, the transient phase establishes a barrier that OSKM cannot overcome and hence leads to lower reprogramming efficiency.

As suggested by the reviewer, we have now included western blots for OCT4 and SOX2 during the differentiation and reprogramming process (**Supplementary Fig. 1c**). As can be seen from these, OCT4 and SOX2 are induced including at the late time points. We also tried KLF4 and cMYC, however, antibodies did not work well for us.

4. Figure 2a. As above, it is not at all clear what Figure 2a depicts. It appears to show that there is a significant increase in colonies obtained following a 72hr pulse versus a 96hr pulse for all but one of the differentiation timepoints assessed. Does this not imply that reprogramming is ongoing at 96hrs?

In **Figure 2a**, the x-axis represents the time duration for which cells are exposed to dox and on y-axis shows what is the normalized fraction of Nanog+ colonies with respect to time point 0. Each colored line shows the fraction of cells, for each pulse duration, that were differentiated for the mentioned time durations i.e. 0, 24, 36, 48, 60 and 72 hours. Taking as an example cells that were differentiated for 24h (yellow line), we observe that when these cells are placed in dox for even 6h, almost 80% of the resultant colonies are Nanog+. After keeping them in dox for 24h, we see a minimal gain as almost all colonies become Nanog+ and therefore any longer dox duration does not change the outcome. However, for cells in the transient phase i.e. cells differentiated for 36h or 48h, we observe a high reprogramming efficiency only if cells are exposed to at least 24h of dox. Hence, these cells which have crossed into the transient phase, require longer OSKM induction to overcome the barrier. Furthermore, once cells cross the transient phase and have committed to differentiation, even a very long OSKM induction of 96h do not lead to higher reprogramming efficiency. The slight increase on Nanog+ signal for cells differentiated for 60h is equivalent to what we would expect for any somatic cell. This is further substantiated by the comparison of AP+ colonies obtained after 4 or 10 days of dox (**Supplementary Figure 1j**) and comparing that with AP+ colonies obtained for MEFs and cells differentiated for 96h (**Supplementary Figure 1i**).

5. Figure 2C. The lack of overlap between Nanog reporter expression and antibody staining is unusual - even in the context of the study by the Padilla-Torres laboratory. The monoallelic expression of Nanog observed in this study is now widely believed to be due to transcriptional bursting of single alleles at very low expression levels and it is unusual to see high protein expression without reporter activity. In Figure 2D as many as 20% of colonies have this pattern. It is especially unusual to see a whole colony being protein positive, reporter negative – occasional, individual cells within a colony may be more expected. Do the authors ever see GFP positive and NANOG negative cells/colonies? Further information about the construction and reliability of the reporter (as above) would be useful, as would further explanation of these unexpected observations.

We believe that lack of overlap between Nanog reporter and antibody is in fact expected as we mainly observe this pattern for cells that are differentiated for at least 48h. Our data show that heterogeneity of reporter activation within single lineages increased as cells were differentiated and reached highest levels at the 48h transition point. After this all iPSC colony forming fates fell sharply, suggesting that the reporter may be particularly sensitive to early induction steps of a deterministic process. Our data also suggests that the locus is first reactivated and then stabilized by downstream mechanisms over the course of several days. Along with low-level expression of NANOG, this pattern of GFP-/NANOG+ could also arise due to sensitivity of antibody stain versus reporter activation. Also, such a pattern can arise due to late activation of Nanog as a result of the difference of half-lives between GFP (>24h) and NANOG (~2h, as estimated by (Abranches et al., 2013)). We do not see instances where a colony is GFP positive but NANOG negative. In the revised version of the manuscript we have provided further details about the cell line.

6. Page 4, line 138-139. A heterogenous transition phase around this timepoint has been shown by (Kalkan et al., 2017). The slight difference in timings can easily be explained by differences in the experimental setup.

When we refer to heterogeneity of transient phase, we are mainly referring to the observation about the heterogenous induction of our reporter GFP which is highest in the cells that are differentiated up to 48h. Although, the population in response to OSKM is showing heterogenous reporter activation, the outcome i.e. emergence of pluripotent iPSC colonies is very homogenous as majority of the cells gave rise to NANOG+ colonies.

7. Oct4 ChIP. How much Oct4 protein is actually present at each timepoint assessed? The abundance of Oct4 protein after 96hrs of differentiation must be quite low?

In the revised version of the manuscript we have provided western blots showing protein levels at each time-point (new **Supplementary Figure 1c**). OCT4 after 96h of differentiation is very low, however, it is still detectable at both mRNA and protein level.

8. In comparing the 'exclusive' versus 'reactivated' Oct4 binding sites, can the authors comment on any underlying differences in the genomic elements bound by Oct4 (promoter versus enhancer regions? Distance to TSS etc). This is quite important when considering why they have on average a higher CpG content, and also why these regions might behave differently during differentiation.

We have provided this information now as new **Supplementary Figures 3d** and **e**. Interestingly, these two set of OCT4 peaks cannot be distinguished based on their genomic distribution or distance to TSS. To further explore any potential differences between these sets we also performed motif enrichment analysis for exclusive, reaccessed and differentiation associated peak sets. This analysis revealed an interesting motif configuration for these regions. As expected, the exclusive peaks show enrichment for usual pluripotency associated transcription factors such as Nanog, Oct4, Sox2 while the differentiation associated peaks show enrichment for TFs that are required for different lineage diversions e.g STAT3, EGR, AP1, SP1 etc. Surprisingly, reaccessed peak set is found to be enriched in TFs from both exclusive as well as differentiation peaks (new

Supplementary Figures 4 and b). The presence of TF motifs from both sets might be partially responsible for the ability of these regions to stay competent for OCT4 binding even after 48h of differentiation. Furthermore, a position specific motif analysis with respect to the peak center also showed a slight divergence of motif location with respect to peak center. While exclusive peaks showed highest preference for the motif at the peak center, reaccessed peaks showed a shift of around 20bp for the best motif site. This flexibility in motif frequency at “reaccessed” sites might also reflect a potential role of co-factors at these regions. Overall our motif analysis indicates that, unlike exclusive regions, reaccessed regions have potential TF binding motifs that can take over once the pluripotency associated TFs are downregulated. The new results are described on pages 5 and 6.

9. Page 4, Line 168. The ‘global reorganization’ the authors discussed is likely due to the high level of heterogeneity at this stage – Oct4 is bound to different places in the different cell types present.

The global reorganization may not be due to heterogeneity in the system as we also observe a global change in overall transcriptome (**Figure 3a**). These changes are likely due to the inherent molecular events that happen when cells commit to a differentiated fate. Changes at 48h are likely associated with specification event during the differentiation process where cells are transcriptionally downregulating many pluripotency associated TFs and hence may lead to reorganization of OCT4 binding as reported in previous studies (Buecker et al., 2014; Yang et al., 2014). At 96h we again see a similar global reorganization which is likely associated with determination event when cells become committed to differentiate.

10. Page 5, line 200. Is it not true by definition that Oct4 would not reaccess the pluripotency exclusive regions? The nomenclature and interpretation becomes extremely confusing here.

Both “exclusive” as well as “reaccessed” regions are bound by OCT in ESCs. However, we termed a subset of all pluripotency associated OCT4 regions “exclusive” based on absence of OCT4 binding during pluripotency reacquisition (**Supplementary Figure 3a**). While the regions that showed OCT4 occupancy during reprogramming were termed “reaccessed”.

11. Page 5, Line 220. A clear description of the somatic tissues used is essential to the interpretation of this data – including why these make comparable and informative data sets to assess alongside the early embryonic data.

The full list of somatic tissues is provided in the methods section of the manuscript including the GEO accession numbers of the sequencing data (page 13). These datasets were selected as several somatic tissues are routinely used in reprogramming experiments, so we wanted to check the DNA methylation status of “reaccessed” regions. Furthermore, we observed that during global remethylation as cells transition from ICM to epiblast “reaccessed” regions did not fully regain the DNA methylation. This prompted us to explore if these regions stay relatively hypomethylated even in somatic tissues.

12. Page 5, Line 223-224 It appears the numerical data is inverted.

We thank the reviewer for pointing this out. We have corrected the numerical data.

13. Page 6, Line 240 – 270. Many of the conclusions drawn here seem overstated. The data presented is correlative, and is equally consistent with the simplistic notion that sites that are accessible in the early stage of reprogramming are more likely to bind Oct4.

We do not think that open sites are more likely to be bound by OCT4. The accessibility difference between “exclusive” and “reaccessed” in somatic tissues is very minimal as shown in **Supplementary Figure 5e and 5f**. Furthermore, based on data from *D. Li et. al.* where they divided the peaks that become open from closed either during early, middle or late stages of reprogramming, we observed a clear enrichment for the early stage. This further supports our claim that these sites are closed to begin with and then gain accessibility. It has been shown that

presence of co-factors such as SOX2 or KLF4 enable Oct4 cooperatively to act as a pioneer factor and bind closed chromatin regions (Chronis et al., 2017). However, in the absence of co-factors OCT4 tends to occupy open chromatin regions (Donaghey et al., 2018).

14. The discussion needs a substantial re-write to better represent the data presented, its caveats and correlative nature.

Again, we must say that we do not fully agree with the reviewer's assessment that discussion requires a substantial re-write. Results should report facts while conclusions/discussion should be "our" interpretation of the result and they can be in part speculative. We agree that the data showing importance of these regions in reprogramming is of correlative nature and in this study, we aim establish the basic concepts of how specific set of genomic regions interact with transcription factors to regulate the reprogramming process. Further future studies are planned to specifically investigate the role of these regions in reprogramming as well as during differentiation to highlight the importance of the barrier that is established early during the differentiation process.

In particular, but not exclusively:

- Line 273 'involves' – 'correlates with' would be more appropriate

We have changed the text accordingly.

- Line 276 'discrete molecular determination event' – what do the authors envisage this is? Why is it discrete?

As indicated above, the differentiation process can be divided into different stages where the final differentiation event is preceded by the commitment of cells to a certain fate. Commitment is also a staged process which involves specification and determination events. Specification is a state from where cells can revert, however, once cells have undergone determination event they are committed to differentiate. With our experiments and data, we have shown that cells differentiated till 24 hours can revert without any exogenous cues, hence they have reached the state of specification. While after 48 hours cells cannot revert efficiently even when supplemented with exogenous cues as they are committed to differentiate. The transition period where cells cannot revert with just changing media but can do so upon ectopic expression of OSKM, have well demarcated boundaries which corresponds to molecular events including changes in global transcriptome and re-organization of TF binding. Hence, we believe that this is a discrete molecular determination event that demarcates the cells that are fully committed from those that can still revert.

- Line 277 – 'extended latency' – have they shown this?

We believe that in discussion one can extrapolate the observations and their implication in scenarios other than the current system that is being tested. We agree that we have not tested for "extended latency" during somatic reprogramming, however, by the said statement we are extrapolating that such a barrier may subsequently hinder somatic reprogramming and may also be responsible for the observed latency during the process.

- Line 279 'through induction of a core subset of pluripotency genes' – again this was a correlation. This is a hypothesis that could be tested experimentally.

We understand that any global RNA-seq or ChIP-seq analysis can only provide information about which genes or genomic regions are modulated during a process. However, such studies are now well-accepted to claim global trends about the process. In our statement, we are also not claiming any specific genes or regions but only that a set of genes that show differential expression and a set of regions that are differentially bound by OCT4 are likely involved in the process of reprogramming. We fully agree that further experiments are required to specifically say that binding of OCT4 to these regions is essential for reprogramming. However, that doesn't negate the fact that these regions and genes are part of initial changes that occur during the process.

- Line 281 – ‘Oct4s ability to access these regions appears to dictate reprogramming outcome’ – again this is based on correlation, and in this context Oct4 is the only transcription factor for which they have assayed the binding – so it is very hard to draw such a conclusion. Again, there are range of experiments which could be undertaken to test this notion. Of note, the authors have not demonstrated that exogenous Oct4 is required for reprogramming in this system.

Again, we have not made the claim that OCT4 at these regions is essential, but we specifically say that this OCT4 ability “appears to dictate”. Hence, based on ChIP-seq data and correlation with somatic reprogramming, we believe this statement should be acceptable.

We are not very clear why the reviewer is stating that we have not shown that exogenous OCT4 is required for reprogramming in this system. For all time-points we have without dox condition where OCT4 is naturally depleted and in those conditions cells after 48h of differentiation do not show NANOG+ colonies. Therefore, we have presented several observations that indicate that exogenous OCT4 by dox induction is required in our system for reprogramming.

- Line 295 ‘persistent epigenetic artefact’ – this is a bizarre concept. It seems unlikely that development leaves artefacts to aid future attempts at induced pluripotency

We do not intend to say that these regions show remnants of pluripotent epigenetic signals for any future attempts of induced pluripotency. There could be different reasons why they escape global remethylation. Given that a set of pluripotency associated transcriptional and epigenetic program is utilized for PGC development (Kehler et al., 2004; Li et al., 2017; Tang et al., 2015; Tang et al., 2016) (Respuela P, Curr. Stem Cell Rep, 2016), we speculate that these regions could be important for PGC development. Hence, to keep them active for PGC development these regions are kept protected and hence escape remethylation. Once the remethylation wave is over they remain so and are passed as such to somatic tissues. The opposite of such memory is also observed during the reprogramming process where iPSC cells show either remnants of their somatic source or acquire epigenetics features that are hallmark of their somatic source (Beagan et al., 2016; Krijger et al., 2016; Nefzger et al., 2017).

- Line 296-297: ‘residual signature of the pluripotent state is analogous to the proposed epigenetic memory of somatic patterning retained in iPSCs derived from different cell types’. This extends the unusual suggestion above. By what criteria are these regions classified as part of a ‘residual signature of the pluripotent state’.

We have observed that they retain a DNAm profile, which is similar to ESCs as they do not get remethylated while the “exclusive” regions are fully methylated in epiblast and in somatic tissues. Based on these observations, we believe that “reaccessed” regions show a residual epigenetic signature from their pluripotent state.

- Line 306-310: the conclusions drawn here are not backed up by the data presented.

These are our conclusions/interpretations and readers can ignore these and judge themselves based on the results. Therefore, we do not believe that the main points of the manuscript as stated in the final sentences of the manuscript are not supported by our experiments. Based on our population based NANOG reporter assays and live cell imaging we believe we were able to pinpoint the two points of separations during which differentiating cells can or cannot revert to pluripotency. We would like to convey that experiments and data shown in **Figure 1, 2** and **Supplementary Figure 1** fully support the notion that cells differentiated to less than 24h can reprogram with high efficiency while once they go beyond 48h reprogramming efficiency drops to somatic levels.

15. Referencing – the literature on exit from pluripotency should be cited. In addition, the reference 24 (Page 4, Line 153) appears to be a miscitation. The referenced paper studies the transition between pluripotent states, rather than exit of pluripotency.

We thank the reviewer for indicating the relevant citations and apologize for missing them. We have included the said citations in the current version on page 2.

References:

Hatano, S.-Y., Tada, M., Kimura, H., Yamaguchi, S., Kono, T., Nakano, T., Suemori, H., Nakatsuji, N., and Tada, T. (2005). Pluripotential competence of cells associated with Nanog activity. *Mech Dev* 122, 67–79.

Kalkan, T., Olova, N., Roode, M., Mulas, C., Lee, H.J., Nett, I., Marks, H., Walker, R., Stunnenberg, H.G., Lilley, K.S., et al. (2017). Tracking the embryonic stem cell transition from ground state pluripotency. *Development* 144, 1221–1234.

Mulas, C., Kalkan, T., and Smith, A. (2017). NODAL Secures Pluripotency upon Embryonic Stem Cell Progression from the Ground State. *Stem Cell Reports* 9, 77–91.

Reviewer #3 (Remarks to the Author):

In this manuscript, Sudhir Thakurela and colleagues optimized an experimental system, in which pluripotent cells were first differentiated and then reinduced into pluripotency. With this approach, the authors found a transient period before the re-establishment of pluripotency, and defined some OCT4 bounding sites with distinct epigenetic signature. This is a novel approach and will be useful for the field. However, the authors should provide some insights on how to improve our understanding to the field, thus some further work may be needed before publication.

We thank the reviewer for encouraging comments on our manuscript. We have provided further details and answers to the specific questions raised by the reviewer.

Major concerns:

1 The authors found that, “cells differentiated beyond the proposed transient phase can still reprogram, but the efficiency and time required resembles those observed for mouse embryonic fibroblasts (MEFs)”. Based on this discovery, the authors infer that ,a barrier similar to the one in somatic cell reprogramming is imposed shortly after exit from the pluripotent state. This is quite interesting. It would be great if the authors can find out what exact the barrier is and to prove the “similarity” by experiments?

We thank the reviewer for finding our observations interesting and the suggestion of showing similarities on this barrier during reprogramming and differentiation. We believe a feature of the barrier is OCT4 access to certain sites during the transient period that makes the cells reprogram with high efficiency. In our experiments we have already shown that cells during the transient phase reprogram with high efficiency. Pinpointing exactly all the molecular parts of the barrier goes beyond what we can currently accomplish. We did a few additional experiments, which we provide for the reviewers but don't feel they would currently improve the manuscript. This might need more work in the future. Briefly, to further explore the similarity between reprogramming and differentiation, we asked if the transient state cells have higher differentiation potential. To test this, we induced differentiating cells to commit to either the mesodermal (ME) lineage using CHIR or neuroectodermal (NE) lineage by inhibition of FGF, Nodal, and BMP signaling. Efficiency of ME or NE lineage induction was scored by BRACHYURY (T) or SOX1 immunofluorescence, respectively, and demonstrates maximal relative induction efficiency towards both ME and NE fates when exogenous signaling cues are added to differentiating cells within the transient phase (**Reviewer Figure 1**). The relative peak in fate induction potential, at approximately 60h following LIF/2i withdrawal, coincides temporally with passage from the transient to the differentiated phase, which occurs at the time when cells acquire a somatic equivalent robustness to perturbation in response to either exogenous signaling or endogenous TF over expression. This is preliminary data that conveys that the transient phase has higher reprogramming as well as differentiation potential and indicates that the barrier that is established after the transient phase

impedes forward as well as reverse potential of differentiating cells. We understand that further validation and molecular exploration is required to pinpoint the molecular events that define the barrier. Future studies should aim to explore this barrier further, however, we believe that our current observations about this transition are important enough to be shared with the scientific community using an open access platform like Nature Communications.

Reviewer Figure 1: Representative images of cells directed to differentiate towards either the mesendodermal (top row) or neuroectodermal (bottom row) lineage, immunostained for brachyury (T) or Sox1, respectively. Scale bar = 200 μm. Heatmaps underneath the images indicate efficiency of lineage induction, scaled to the measured maximum and minimum range of T or Sox1 signal according to lineage.

2 The authors performed Oct4-ChIP-seq, and defined several sets of Oct4 targets, with a subset one termed “pluripotent reaccessed”, that remain competent for ectopic binding until at least 48h of differentiation. The reviewer wants to know whether the “pluripotent reaccessed” binding sites really contribute to the barrier(s) for the somatic cell reprogramming, it's a consequence or a cause for the somatic cell reprogramming, any key gene or target was response to this process functionally?

We totally agree with the reviewer that currently we have only shown that these regions are required to be bound by OCT4 during in our system and that they are preferentially occupied and show increased accessibility during somatic reprogramming. This study is putting in place the foundation of TF and epigenome interactions during high efficiency reprogramming and highlight how certain genomic regions interact with TFs to facilitate high-efficiency reprogramming. Future studies should explore these regions in greater detail to establish their actual role during the high efficiency reprogramming. It would be interesting to systematically dissect the molecular events that govern access to these regions by performing large scale CRISPR-mediated knock-out of these regions to establish their function not only during reprogramming but also during the differentiation process. This will further establish the similarity of the barrier that is established early during differentiation and hinder high efficiency reprogramming. Furthermore, to specifically identify the co-factors enriched at these regions, it would be interesting to perform a locus-specific pull-down followed by mass-spectroscopy to identify the co-factors that may be involved in keeping these regions accessible in comparison to “exclusive” regions. We hope that reviewer will also understand that a proper cause and consequence dissection of these regions will be out of scope

for this manuscript as it will require an extensive study with primary aim focusing on showing importance of these regions during somatic reprogramming and differentiation.

3 The authors found the accessible peaks exhibit a 2.4–fold higher CpG density as compared to the genomic average. Can this be tested functionally to establish the relationship between them and the barriers for somatic cell reprogramming.

We thank the reviewer for this suggestion, however, we think experimentally testing this hypothesis would again be out of scope for this manuscript as it would require to systematically mutate CpG density in these regions and then perform these assays to see if OCT4 binding is affected or not. With the current study we are just proposing that regions that retain OCT4 occupancy longer during differentiation are more likely to have higher CpG density. We understand that future studies should be planned to systematically dissect out the cause and consequences of TF access to these regions along with the actual role of CpG density in regulating TF binding at these regions. We hope that reviewer will agree with us in the merit of exploring this suggestion in the follow up studies in greater details and also that scientific community will benefit from sharing our current observations in a timely manner.

4 The authors checked the binding of OCT4 by Chip-seq, how about the binding pattern of Klf4 and Sox2?

We thank the reviewer for this suggestion. For the revised manuscript we performed SOX2 ChIP-seq and data and observations can be found in (**new Supplementary Figures 4d-i**). In brief, we observe that SOX2 also follows OCT4 like dynamics where certain regions retain SOX2 binding till the transient phase and then this binding is lost coinciding with loss of high efficiency reprogramming. There is a very high overlap between the OCT4 and SOX2 bound regions that are specifically regained in 48 +dox conditions (**new Supplementary Figure 4f**). As the observation for OCT4 are corroborated by SOX2, we focused the further detailed analysis of only OCT4 sites.

5 The whole story were mainly focus on the description of the data, and discovery of some specific Oct4 accessible regions etc. Further work may be needed to validate some of those findings.

We agree with the reviewer that all the points raised are valid and interesting. If we had the intention to publish the findings in *Nature* we would have certainly attempted to do all of that but feel that the manuscript as is has enough content to be shared with the scientific community using an open access, general interest platform such as *Nature Communications*. Sharing these observations with the broader scientific community will be beneficial in general as others can replicate and extend on these observations.

REFERENCES:

- Abranches, E., Bekman, E., and Henrique, D. (2013). Generation and characterization of a novel mouse embryonic stem cell line with a dynamic reporter of Nanog expression. *PLoS One* 8, e59928.
- Beagan, J.A., Gilgenast, T.G., Kim, J., Plona, Z., Norton, H.K., Hu, G., Hsu, S.C., Shields, E.J., Lyu, X., Apostolou, E., *et al.* (2016). Local Genome Topology Can Exhibit an Incompletely Rewired 3D-Folding State during Somatic Cell Reprogramming. *Cell Stem Cell* 18, 611-624.
- Buecker, C., Srinivasan, R., Wu, Z., Calo, E., Acampora, D., Faial, T., Simeone, A., Tan, M., Swigut, T., and Wysocka, J. (2014). Reorganization of enhancer patterns in transition from naive to primed pluripotency. *Cell Stem Cell* 14, 838-853.
- Buganim, Y., Faddah, D.A., Cheng, A.W., Itskovich, E., Markoulaki, S., Ganz, K., Klemm, S.L., van Oudenaarden, A., and Jaenisch, R. (2012). Single-cell expression analyses during cellular reprogramming reveal an early stochastic and a late hierarchic phase. *Cell* 150, 1209-1222.

Chronis, C., Fiziev, P., Papp, B., Butz, S., Bonora, G., Sabri, S., Ernst, J., and Plath, K. (2017). Cooperative Binding of Transcription Factors Orchestrates Reprogramming. *Cell* 168, 442-459 e420.

Donaghey, J., Thakurela, S., Charlton, J., Chen, J.S., Smith, Z.D., Gu, H., Pop, R., Clement, K., Stamenova, E.K., Karnik, R., *et al.* (2018). Genetic determinants and epigenetic effects of pioneer-factor occupancy. *Nat Genet* 50, 250-258.

Galonska, C., Ziller, M.J., Karnik, R., and Meissner, A. (2015). Ground State Conditions Induce Rapid Reorganization of Core Pluripotency Factor Binding before Global Epigenetic Reprogramming. *Cell Stem Cell* 17, 462-470.

Kalkan, T., Olova, N., Roode, M., Mulas, C., Lee, H.J., Nett, I., Marks, H., Walker, R., Stunnenberg, H.G., Lilley, K.S., *et al.* (2017). Tracking the embryonic stem cell transition from ground state pluripotency. *Development* 144, 1221-1234.

Kehler, J., Tolkunova, E., Koschorz, B., Pesce, M., Gentile, L., Boiani, M., Lomeli, H., Nagy, A., McLaughlin, K.J., Scholer, H.R., *et al.* (2004). Oct4 is required for primordial germ cell survival. *EMBO Rep* 5, 1078-1083.

Krijger, P.H., Di Stefano, B., de Wit, E., Limone, F., van Oevelen, C., de Laat, W., and Graf, T. (2016). Cell-of-Origin-Specific 3D Genome Structure Acquired during Somatic Cell Reprogramming. *Cell Stem Cell* 18, 597-610.

Li, L., Dong, J., Yan, L., Yong, J., Liu, X., Hu, Y., Fan, X., Wu, X., Guo, H., Wang, X., *et al.* (2017). Single-Cell RNA-Seq Analysis Maps Development of Human Germline Cells and Gonadal Niche Interactions. *Cell Stem Cell* 20, 891-892.

Liu, L.L., Brumbaugh, J., Bar-Nur, O., Smith, Z., Stadtfeld, M., Meissner, A., Hochedlinger, K., and Michor, F. (2016). Probabilistic Modeling of Reprogramming to Induced Pluripotent Stem Cells. *Cell Rep* 17, 3395-3406.

Nefzger, C.M., Rossello, F.J., Chen, J., Liu, X., Knaupp, A.S., Firas, J., Paynter, J.M., Pflueger, J., Buckberry, S., Lim, S.M., *et al.* (2017). Cell Type of Origin Dictates the Route to Pluripotency. *Cell Rep* 21, 2649-2660.

Silva, J., Barrandon, O., Nichols, J., Kawaguchi, J., Theunissen, T.W., and Smith, A. (2008). Promotion of reprogramming to ground state pluripotency by signal inhibition. *PLoS Biol* 6, e253.

Tang, W.W., Dietmann, S., Irie, N., Leitch, H.G., Floros, V.I., Bradshaw, C.R., Hackett, J.A., Chinnery, P.F., and Surani, M.A. (2015). A Unique Gene Regulatory Network Resets the Human Germline Epigenome for Development. *Cell* 161, 1453-1467.

Tang, W.W., Kobayashi, T., Irie, N., Dietmann, S., and Surani, M.A. (2016). Specification and epigenetic programming of the human germ line. *Nat Rev Genet* 17, 585-600.

Wernig, M., Meissner, A., Foreman, R., Brambrink, T., Ku, M., Hochedlinger, K., Bernstein, B.E., and Jaenisch, R. (2007). In vitro reprogramming of fibroblasts into a pluripotent ES-cell-like state. *Nature* 448, 318-324.

Yang, S.H., Kalkan, T., Morissroe, C., Marks, H., Stunnenberg, H., Smith, A., and Sharrocks, A.D. (2014). Otx2 and Oct4 drive early enhancer activation during embryonic stem cell transition from naive pluripotency. *Cell Rep* 7, 1968-1981.

Zviran, A., Mor, N., Rais, Y., Gingold, H., Peles, S., Chomsky, E., Viukov, S., Buenrostro, J.D., Scognamiglio, R., Weinberger, L., *et al.* (2019). Deterministic Somatic Cell Reprogramming Involves Continuous Transcriptional Changes Governed by Myc and Epigenetic-Driven Modules. *Cell Stem Cell* 24, 328-341 e329.

Reviewers' comments:

Reviewer #1 (Remarks to the Author):

In this round of revision, Thakurela et al. has answered all my concerns but one. In the original manuscript, I have asked the authors to address the following (point number 5 in the rebuttal letter): "As reprogramming efficiency is affected by OSKM protein levels, the authors need to show how similar the ectopic OSKM protein levels are once induced from the different stages of differentiation."

The authors have indeed looked into this and replied: "We thank the reviewer for the suggestion. We have now included western blots for OCT4 and SOX2, during differentiation and reprogramming (new Supplementary Figure 1c). This suggests proper induction for each time point and protein levels appear comparable. Unfortunately, antibodies for KLF4 and cMYC did not work well for us".

However, I don't agree with the conclusion that Oct4 and Sox2 levels are comparable between 48h and 96h cells. Unless I am grossly misunderstanding what's in Supplementary figure 1c, it is clear that both Oct4 and Sox2 are induced more (darker +dox bands) from cell differentiated for 48h compared to 96h. Even in the absence of Dox (-dox bands), Oct4 levels (either due to the persisting endogenous Oct4 or leaky transgene activation) seem to be higher in 48h cells compared to their 96h counterparts. This may be due to the transgene being more efficiently silenced at 96h and harder to reactivate with dox. Therefore, after 48h of differentiation, a combination of endogenous Oct4 and enhanced induction of ectopic Oct4 results in higher total levels of Oct4 (the same is true for Sox2 and may be true for Klf4 and c-Myc), which will make these cells more prone to reprogramming.

However, I understand that this is confounded by the fact that the western blot was carried 48h after treating with dox, which means that the ectopic Oct4 can be the same between 48h and 96h, if after 48h dox induction, the ectopic Oct4 resulted in more endogenous Oct4 activation compared to 96h. To resolve this issue, the authors may carry out qPCR and measure the levels of OCT4 transcripts (better all OSKM factors) coming from both the endogenous gene and transgene. But in either case, this does not change the fact that the total amounts of Oct4 protein is higher in 48h compared to 96h, which the authors need to clearly point out in the manuscript. Furthermore, this is the time point at which Oct4 and Sox2 ChIP-seq experiments were carried out, which would ultimately have an implication on the interpretation data. I therefore suggest that the authors should consider, as part of their conclusion, that the levels of Oct4 protein as well as the chromatin state are important for Oct4 to bind the reaccessed sites and drive more efficient reprogramming.

Overall, the revised manuscript is much stronger than the original one, but the main conclusions still stand. I believe that this study provide key insights into the iPSC reprogramming process. More importantly, the concept of an epigenetic memory of the pluripotency state is both novel and will draw wide interest beyond the iPSC field. Therefore, I still highly recommend the publication of this manuscript in NATURE COMMUNICATION if the authors address my main concern listed above.

Reviewer #2 (Remarks to the Author):

The revised manuscript has improved in terms of clarity, and in the accuracy of some of the claims. However, in other areas it is disappointing that the authors have chosen to ignore previous comments and suggestions. There remain a number of issues:

1. Experimental system. The transient exposure (24hrs) to 2i/LIF prior to the start of the experiments is unusual. The authors' previous study (Galonska et al. 2015) does not really address my concern

that the cells are likely in flux – in fact it very much agrees that histone modifications have already begun to alter within 24hrs of exposure and that the pluripotency transcription factor network is dramatically changing its configuration. Indeed, the consequence of such treatment immediately prior to a period of differentiation – and whether this does indeed trigger a more ‘homogenous’ differentiation - seem completely open questions. In agreement, the authors state in the manuscript that ‘cells undergo extensive epigenetic remodeling and molecular restructuring in response to 2i’ – a process which has begun and is certainly not complete within a 24hr pulse. At the very least, the short treatment with 2i/LIF should be made explicit in the text of the results section and in Figure 1a – otherwise a casual reader might envisage that this study has been performed with cells stably expanded in 2i/LIF. Beyond this, I am of the opinion that this unusual experimental design may well impact the results obtained (and, to an extent, the authors agree (lines 158-160)). However, I do note that the authors present new evidence that some of their findings, such as the behaviour of reaccessed peaks, may be generalisable to other systems.

2. Reaccessed peaks: In this version of the manuscript the ‘reaccessed’ regions emerge much more obviously as the most novel and interesting aspect. As such, I have some remaining questions and comments regarding these.

- I think the authors could explain more clearly what is distinctive about these peaks. For instance, at the 48hr timepoint (for instance, Fig 4d) are the differences between Exclusive and Reaccessed statistically significant for any of the assays performed (i.e. is the ‘slight delay’ (line 216) simply a trend?). On a related point there seems to be a disconnect in the message between the statement in the introduction ‘these regions retain a distinct epigenetic signature during in vitro and in vivo differentiation’ and the Discussion ‘retention of an open chromatin signature at reprogramming-associated cis-elements is not the sole permissive factor for deterministic reprogramming: while euchromatic signatures persist at these enhancers past the transition point, they are nonetheless initially refractory to OCT4 binding when ectopic OSKM is induced in cells differentiated for more than 48h’. (i.e. is the measured epigenetic signature (not including the DNA methylation re-analysis) of these regions important or not?)

- combining expression and Oct4 binding data, do reaccessed peaks occur in proximity to genes that remain expressed (in comparison Exclusive regions)? And are these genes becoming up or downregulated? Are the associated genes part of a common pathway(s)?

- the addition of the Sox2 ChIP is interesting, but the data could be more clearly/thoroughly explained. For instance, in the comparison shown in Supp Figure 4f, the genes described as reaccessed appear to be all of the peaks in each condition at 48h +dox (almost 12k sites for Oct4), whereas in the rest of the manuscript the term reaccessed sites is used for the ‘pluripotency reaccessed’ Oct4 binding sites which number 3151. What proportion of these 3151 reaccessed sites are co-bound by Sox2? Is there further analysis description that could be interesting here?

- in the last section (lines 274 – 289) the authors introduce another term ‘Oct4 accessible’. This leads to further confusion about exactly which sites are being considered here? (is this the 3151 sites?). Which sites are being considered will impact the interpretation of the findings.

- the idea that these peaks might be utilized during PGC development (as suggested by the authors in their rebuttal) is an interesting one and worthy of inclusion in the Discussion, especially given the literature on reaccessing pluripotency in the germ line.

3. Heterogeneity: The authors use of heterogeneity/homogeneity is unhelpful and does not accurately describe their findings. To describe the process as ‘homogenous’ based solely on its high efficiency does not make sense. With regards the authors explanation of their usage, I would comment that people homogeneously die, but that the cause, timing and mechanism(s) can be quite diverse. The emergence of colonies certainly does not look homogenous, and indeed there are differences in colony size, reporter activity and Nanog protein expression. This manuscript does not deal with, or overcome the ‘heterogeneity problem’ in any meaningful way. Such claims should be removed from the

manuscript, in particular line 65 – 68. Indeed, the large error bars in Figure 1d likely reflect the temporal heterogeneity observed in differentiation.

4. Nomenclature:

- Transition point/state. What does 'estimated the transition points' mean? This does not seem to be clearly enough investigated/defined to then move on to describing the greyed out areas in Figure 1d as a 'transition state' (line 127)
- the justification for the use of the term 'switch' or 'switch-like' remains weak. It implies a rapid homogenous transition, which the authors have not shown. Rather, they have defined a time window during which reprogramming can occur with high efficiency. I would recommend removing this terminology from the manuscript.
- line 128-129 'swift and homogeneously acting mechanisms'. The authors have shown no evidence that the reprogramming process is necessarily swift or homogenous (see above also). Delete.
- line 145-149. 'a rapid, coordinated switch (as the majority of cells reprogram)'. The explanation in brackets does not provide any meaningful explanation. The authors have not shown the process is rapid or coordinated – and the fact that the majority of cells reprogram is irrelevant with regards these terms. Notably in the next statement the authors explain that the process is heterogenous with regards the major marker they use to monitor the process. As mentioned previously, these claims should be removed.

Taken together this language is not helpful and does not accurately describe the complex cellular transitions that are occurring in the differentiation/reprogramming assay used. The authors would be better served by more simply describing what the experiments actually demonstrate. The manuscript is much improved by the removal of the claims regarding 'bistability' and would be further improved by removing the terms above.

5. Claims & Discussion

- Line 269 'unique epigenetic dynamics in our model' – Related to point 2. What exactly do the authors mean here? Delayed loss of ATAC-seq signal? Also, is the 'epigenetic memory' referred to in the next line simply referring to the measured difference in DNA methylation in somatic cells?
- Line 309 'discrete molecular determination event' – the authors defence of this description is not satisfactory. While they do show the basic timing of when determination/commitment has occurred, they do not really demonstrate it is a 'discrete event'.
- line 314. 'appears to dictate'. There is no evidence for this claim. I would suggest 'correlates with'. In their rebuttal the authors claim that they have shown that exogenous OCT4 is required for reprogramming. To be clear, they've shown that exogenous OSKM is required. In their experimental setup it is possible that SKM (or a combination of different factors) might be able to induce reprogramming – especially as Oct4 protein appears to be present for quite some time even in the - Dox condition (Supp Figure 1).
- line 326 'delayed silencing during early differentiation' – what exactly does this refer to? Please clarify and use a more specific description.
- line 331-2 – 'distinct and persistent epigenetic state in fully differentiated adult tissues' – this does seem to be a slight overstatement. The authors have only really shown a DNA methylation difference in adult tissues. Otherwise, it remains possible that the epigenetic state may actually be quite similar. In addition, in the Abstract a similar claim ('epigenetic signature during... in vivo differentiation') is made: this also seems too general a statement, particularly for inclusion in the Abstract.
- Line 342-346. I agree that the authors' data is interesting in the context of cell state transitions. In my original review I suggested the authors put their work in the context of two recent papers from Austin Smith's lab, which the authors give brief mention to in the introduction. In fact, the Discussion would be the ideal place to comment further on this. In particular how their 'discrete moments of specification and determination' might map onto the cell states described by Smith as naïve, formative (Smith, Development 2017) and primed – which would appear to be extremely relevant here.

Minor comments:

- the image quality in Figure 2c may not be suitable for publication.
- Line 174 – 176 'shows limited binding to a minimal set of constitutively bound regions with overall low enrichment'- this statement/description is difficult to understand. Please reword or add further explanation of the pattern binding pattern.

Reviewer #3 (Remarks to the Author):

All revisions are satisfactory. This is ready for publication

We would like to thank all three reviewers for the additional feedback and time invested into further improving the study. Below we provide a response to the remaining questions and hope the manuscript is now acceptable for publication in *Nature Communication*.

Reviewer #1 (Remarks to the Author):

In this round of revision, Thakurela et al. has answered all my concerns but one. In the original manuscript, I have asked the authors to address the following (point number 5 in the rebuttal letter): “As reprogramming efficiency is affected by OSKM protein levels, the authors need to show how similar the ectopic OSKM protein levels are once induced from the different stages of differentiation.” The authors have indeed looked into this and replied: “We thank the reviewer for the suggestion. We have now included western blots for OCT4 and SOX2, during differentiation and reprogramming (new Supplementary Figure 1c). This suggests proper induction for each time point and protein levels appear comparable. Unfortunately, antibodies for KLF4 and cMYC did not work well for us”. However, I don’t agree with the conclusion that Oct4 and Sox2 levels are comparable between 48h and 96h cells. Unless I am grossly misunderstanding what’s in Supplementary figure 1c, it is clear that both Oct4 and Sox2 are induced more (darker +dox bands) from cell differentiated for 48h compared to 96h. Even in the absence of Dox (-dox bands), Oct4 levels (either due to the persisting endogenous Oct4 or leaky transgene activation) seem to be higher in 48h cells compared to their 96h counterparts. This may be due to the transgene being more efficiently silenced at 96h and harder to reactivate with dox. Therefore, after 48h of differentiation, a combination of endogenous Oct4 and enhanced induction of ectopic Oct4 results in higher total levels of Oct4 (the same is true for Sox2 and may be true for Klf4 and c-Myc), which will make these cells more prone to reprogramming. However, I understand that this is confounded by the fact that the western blot was carried 48h after treating with dox, which means that the ectopic Oct4 can be the same between 48h and 96h, if after 48h dox induction, the ectopic Oct4 resulted in more endogenous Oct4 activation compared to 96h.

We agree with the reviewer that these are important considerations.

To resolve this issue, the authors may carry out qPCR and measure the levels of OCT4 transcripts (better all OSKM factors) coming from both the endogenous gene and transgene.

Unfortunately, the endogenous and ectopic factors cannot be distinguished in our system.

But in either case, this does not change the fact that the total amounts of Oct4 protein is higher in 48h compared to 96h, which the authors need to clearly point out in the manuscript. Furthermore, this is the time point at which Oct4 and Sox2 ChIP-seq experiments were carried out, which would ultimately have an implication on the interpretation data. I therefore suggest that the authors should consider, as part of their conclusion, that the levels of Oct4 protein as well as the chromatin state are important for Oct4 to bind the reaccessed sites and drive more efficient reprogramming.

We thank the reviewer for the suggestion and have updated the text on page 8 (Discussion) in the following manner:

Finally, it is worth noting that the retention of an open chromatin signature at reprogramming-associated *cis*-elements is not the sole permissive factor for deterministic reprogramming: while euchromatic signatures persist at these regions past the transition point, they are nonetheless initially refractory to OCT4 binding when ectopic OSKM is induced in cells differentiated for more than 48h. One factor in this context may be the decreasing **total OCT4 levels (endogenous plus ectopic) during the differentiation, which may affect pluripotency reacquisition along with the epigenetic state of re-accessed targets.**

Overall, the revised manuscript is much stronger than the original one, but the main conclusions still stand. I believe that this study provide key insights into the iPSC reprogramming process. More importantly, the concept of an epigenetic memory of the pluripotency state is both novel and will draw wide interest beyond the iPSC field. Therefore, I still highly recommend the publication of this manuscript in NATURE COMMUNICATION if the authors address my main concern listed above.

We want to again thank the reviewer for the careful reading and thoughtful feedback throughout the review process.

Reviewer #2 (Remarks to the Author):

The revised manuscript has improved in terms of clarity, and in the accuracy of some of the claims. However, in other areas it is disappointing that the authors have chosen to ignore previous comments and suggestions. There remain a number of issues:

1. Experimental system. The transient exposure (24hrs) to 2i/LIF prior to the start of the experiments is unusual. The authors' previous study (Galonska et al. 2015) does not really address my concern that the cells are likely in flux – in fact it very much agrees that histone modifications have already begun to alter within 24hrs of exposure and that the pluripotency transcription factor network is dramatically changing its configuration. Indeed, the consequence of such treatment immediately prior to a period of differentiation – and whether this does indeed trigger a more 'homogenous' differentiation - seem completely open questions. In agreement, the authors state in the manuscript that 'cells undergo extensive epigenetic remodeling and molecular restructuring in response to 2i' – a process which has begun and is certainly not complete within a 24hr pulse. **At the very least, the short treatment with 2i/LIF should be made explicit in the text of the results section and in Figure 1a** – otherwise a casual reader might envisage that this study has been performed with cells stably expanded in 2i/LIF. Beyond this, I am of the opinion that this unusual experimental design may well impact the results obtained (and, to an extent, the authors agree (lines 158-160)). However, I do note that the authors present new evidence that some of their findings, such as the behaviour of reaccessed peaks, may be generalisable to other systems.

We thank the reviewer for continuing this thoughtful review aimed at improving the manuscript. As suggested by the reviewer to explicitly state the 24h pre-treatment, we have now included this in the manuscript text as well in Figure 1a (Results; Paragraph 1; page 2):

“Cells were cultured in Serum/LIF, **exposed to 2i/LIF media for 24h and then allowed** to differentiate by switching into N2B27 media (Fig. 1a).”

“2. Reaccessed peaks: In this version of the manuscript the 'reaccessed' regions emerge much more obviously as the most novel and interesting aspect. As such, I have some remaining questions and comments regarding these.

- I think the authors could explain more clearly what is distinctive about these peaks. For instance, at the 48hr timepoint (for instance, Fig 4d) are the differences between Exclusive and Reaccessed statistically significant for any of the assays performed (i.e. is the 'slight delay' (line 216) simply a trend?). On a related point there seems to be a disconnect in the message between the statement in the introduction 'these regions retain a distinct epigenetic signature during in vitro and in vivo differentiation' and the Discussion 'retention of an open chromatin signature at reprogramming-associated cis-elements is not the sole permissive factor for deterministic reprogramming: while euchromatic signatures persist at these

enhancers past the transition point, they are nonetheless initially refractory to OCT4 binding when ectopic OSKM is induced in cells differentiated for more than 48h'. (i.e. is the measured epigenetic signature (not including the DNA methylation re-analysis) of these regions important or not?)

As requested by the reviewer, we now provide a new plot with the statistical significance. As shown in Supplementary **Fig3b** (NEW FIGURE, right panel), reaccessed regions at 48h and 96h show significantly higher enrichment for H3K4me2, H3K27ac, and ATAC-seq as compared to exclusive regions in respective “-dox” as well as +dox” conditions. Furthermore, when compared to 96h+dox, reaccessed regions in 48+dox conditions gain significantly more H3K27ac, H3K4me3 and ATAC signal. Although reaccessed regions show significant gain even at 96h, this enrichment is several orders of magnitude lower when compared to 48h time-points. Taken together along with Fig. 4d, reaccessed regions appear to retain euchromatic signature until 96h and they significantly gain H3K27ac, H3K4me2 and ATAC-seq when cells are induced with OSKM. Relevant text is added at page 5:

“Furthermore, during pluripotency reacquisition after 48h of differentiation (+ dox), reaccessed sites **significantly gain H3K27ac, H3K4me2, and ATAC-seq signal, supporting a functional response to ectopic OCT4 binding** (Supplementary Figs. 3b, 3f, 3g).”

We would like to clarify that in the abstract when we say, “these regions retain a distinct epigenetic signature during in vitro and in vivo differentiation”, we refer to reduced DNAm at these regions which is carried over to somatic tissues as they show incomplete re-methylation during early development. Alternatively, in the discussion we refer to our observations about the H3K27ac, H3K4me2, and ATAC-seq signal over these regions post 48h. Although we show that these regions maintain a relatively active chromatin structure at 96h as compared to exclusive regions, it appears to be not enough to facilitate OCT4 binding at 96h. This may either be due to the lack of additional co-factors or accessibility has decreased to an extent that is no longer supporting transcription factor binding. And as we show in Fig. 5, with further downstream differentiation the re-accessed regions are also shut-down completely as shown by the DNase hypersensitivity data from somatic tissues. Therefore, we believe that the differential epigenetic state of re-accessed regions facilitates OCT4 binding at 48h and consequently these regions maintain a relatively stronger euchromatic structure even at 96h, however, it is not sufficient to facilitate TF binding.

- combining expression and Oct4 binding data, do reaccessed peaks occur in proximity to genes that remain expressed (in comparison Exclusive regions)? And are these genes becoming up or downregulated? Are the associated genes part of a common pathway(s)?

The reaccessed peaks are indeed enriched for genes that are up-regulated upon dox induction. As reported in Fig. 4f, genes in proximity to the reaccessed peaks are enriched for the set of genes that are induced at 48h +dox. Furthermore, it is noteworthy that these genes do not respond in similar way at 96h as we observe induction of genes associated with differentiation peaks.

In response to the reviewer suggestion, we have now performed new enrichment analysis for genes that are induced and are close to peaks associated with either the exclusive or reaccessed set. This analysis (**Supplementary Fig. 3h; NEW FIGURE**) indicates a significant enrichment for biological processes related to “stem cell population maintenance” and “embryo development” or more general early developmental. We have added a sentence in the main text on page 5. The added sentence is referred below:

“The set of reaccessed peaks are also more frequently proximal to genes that are transcriptionally upregulated following OSKM induction after 48h of differentiation **and these genes are involved in processes related to stem cell maintenance and early development.**”

- the addition of the Sox2 ChIP is interesting, but the data could be more clearly/thoroughly explained. For instance, in the comparison shown in Supp Figure 4f, the genes described as reaccessed appear to be all of the peaks in each condition at 48h +dox (almost 12k sites for Oct4), whereas in the rest of the manuscript the term reaccessed sites is used for the 'pluripotency reaccessed' Oct4 binding sites which number 3151. What proportion of these 3151 reaccessed sites are co-bound by Sox2? Is there further analysis description that could be interesting here?

First, we apologize that the number 3,151 was a typographical mistake as the number of reaccessed peaks is 3,550. We have corrected this in the revised manuscript. As to the reviewer's question, out of these 3550 reaccessed peaks, a stringent RPKM cutoff for SOX2 enrichment (RPKM>2) indicate that 2,496 are also bound by SOX2 in 48h +dox condition. Relaxing the SOX2 enrichment to 1.5 shows that 3,112 peaks are bound by SOX2 in 48+dox condition. We have explored SOX2 peaks independently and they appear to show similar features like OCT4 and considering that OCT4 reaccessed peaks are also bound by SOX2 and overall SOX2 peaks at 48+dox are highly overlapping with OCT4, we decided to keep the focus of the study to OCT4 peaks. To clearly specify the overlap of OCT4 re-accessed peaks with SOX2 at 48+dox, we have updated the text with the following sentence on page 6:

“Specifically, out of 3,550 OCT4 reaccessed peaks, 2,496 also show high SOX2 enrichment (RPKM>2) and reducing the enrichment cutoff further (RPKM>1.5) increases the overlapping SOX2 binding set to 3,112 peaks. Given this high overlap between OCT4 and SOX2 binding, we subsequently focused our subsequent analysis on OCT4.”

- in the last section (lines 274 – 289) the authors introduce another term 'Oct4 accessible'. This leads to further confusion about exactly which sites are being considered here? (is this the 3151 sites?). Which sites are being considered will impact the interpretation of the findings.

We apologize for the confusion as we are indeed referring to OCT4-reaccessed peaks. We have made appropriate changes to specifically refer to reaccessed peak set throughout the manuscript.

- the idea that these peaks might be utilized during PGC development (as suggested by the authors in their rebuttal) is an interesting one and worthy of inclusion in the Discussion, especially given the literature on reaccessing pluripotency in the germ line.

We thank the reviewer for this suggestion and have now included this in the discussion at page 8:

The transition state in our system emerges as the pluripotency network is dismantled but precedes establishment of a differentiated cellular identity. As such, it aligns principally with the recently described “formative pluripotency” stage that separates naïve pluripotency from lineage specification during gastrulation^{23,39}. **Part of the pluripotency associated transcriptional and epigenetic program is also utilized during primordial germ cell (PGC) development⁴⁰⁻⁴³ leading us to speculate that these regions could be important for PGC development. In this model, preserving germline potential within the pluripotent epiblast would require protection of critical regions during global genome remethylation, leading to a persistent epigenetic memory that is carried into rest of the soma.** This residual signature of the pluripotent state appears analogous to the proposed epigenetic memory of somatic patterning retained in iPSCs derived from different cell types⁴⁴⁻⁴⁶.

3. Heterogeneity: The authors use of heterogeneity/homogeneity is unhelpful and does not accurately describe their findings. To describe the process as 'homogenous' based solely on its high efficiency does not make sense. With regards the authors explanation of their usage, I would comment that people homogenously die, but that the cause, timing and mechanism(s) can be quite diverse. The emergence of colonies certainly does not look homogenous, and indeed there are differences in colony size, reporter activity and Nanog protein expression. This manuscript does not deal with, or

overcome the ‘heterogeneity problem’ in any meaningful way. Such claims should be removed from the manuscript, in particular line 65 – 68. Indeed, the large error bars in Figure 1d likely reflect the temporal heterogeneity observed in differentiation.

We have edited the manuscript and have made appropriate changes as requested by the reviewer. The introduction has also been updated on page 1:

“To complement these prior studies, we designed an experimental approach that challenges differentiating pluripotent cells to reacquire their original state under distinct conditions.”

4. Nomenclature:

- Transition point/state. What does ‘estimated the transition points’ mean? This does not seem to be clearly enough investigated/defined to then move on to describing the greyed out areas in Figure 1d as a ‘transition state’ (line 127)

With transition point we refer to approximate time point when cells transition from high to low efficiency reprogramming. We have now added appropriate introductory line about transition point and have also indicated that details are provided in the Methods section. We thank the reviewer for pointing this out. The edited line is on page 3:

“To specifically define the timepoint when cells transition from high- to low-efficiency, we fitted sigmoid curves to the reversion efficiencies of both conditions at each time-point and estimated their respective transition points (see Methods for details, Fig. 1d). “

- the justification for the use of the term ‘switch’ or ‘switch-like’ remains weak. It implies a rapid homogenous transition, which the authors have not shown. Rather, they have defined a time window during which reprogramming can occur with high efficiency. I would recommend removing this terminology from the manuscript.

We have now edited the manuscript text to reflect the requested changes. The changes we made are at the following places:

- Page 3, Section Title: Section title is now changed to **“Transition state cells are uniquely amenable to high efficiency reprogramming”**

- Page 3, Para 2, Line 1: Sentence now reads as: **“The striking resistance to OSKM-induction that is acquired after 48h suggests the existence of a phase during differentiation where cells become permanently committed to a NANOG⁻ (differentiated) state.”**

-Page 4, line 1: Sentence now reads as: **“The frequency of successfully reprogrammed lineages over differentiation reveals a similar transition behavior between high and low efficiency reprogramming states to the ones we observed in our static population-level assays (Supplementary Figs. 1l ,1m).”**

- line 128-129 ‘swift and homogeneously acting mechanisms’. The authors have shown no evidence that the reprogramming process is necessarily swift or homogenous (see above also). Delete.

We have made the appropriate changes in the manuscript text and have tried to convey the message without the use of specific terminology. These changes were made on page 3: Sentence now reads as: **“Taken together, these results suggest that prior to the transition state a minimal duration of dox is needed to direct cells back into the pluripotent state, which is subsequently consolidated by transgene independent mechanisms.”**

- line 145-149. ‘a rapid, coordinated switch (as the majority of cells reprogram)’. The explanation in brackets does not provide any meaningful explanation. The authors have not shown the process is rapid or coordinated – and the fact that the majority of cells reprogram is irrelevant with regards these terms. Notably in the next statement the authors explain that the process is heterogeneous with regards the major marker they use to monitor the process. As mentioned previously, these claims should be removed.

Taken together this language is not helpful and does not accurately describe the complex cellular transitions that are occurring in the differentiation/reprogramming assay used. The authors would be better served by more simply describing what the experiments actually demonstrate. The manuscript is much improved by the removal of the claims regarding ‘bistability’ and would be further improved by removing the terms above.

We thank the reviewer for making these suggestions. We have now edited these sentences on page 4:

“Overall, our imaging-based lineage tracing indicates a **shift** from NANOG⁻ to NANOG⁺ states that is most heterogeneous, in terms of reporter activation, during the transition phase (24 to 48h) before cells go from high to low OSKM induced reprogramming efficiency.”

5. Claims & Discussion

- Line 269 ‘unique epigenetic dynamics in our model’ – Related to point 2. What exactly do the authors mean here? Delayed loss of ATAC-seq signal? Also, is the ‘epigenetic memory’ referred to in the next line simply referring to the measured difference in DNA methylation in somatic cells?

In this statement we not only refer to ATAC-seq signal but also H3K27ac and H3K4me2 signal at 48 and 96h. Reaccessed peaks show delayed erasure of these features and hence here we collectively refer to them as unique epigenetic dynamics. And by epigenetic memory, we are indeed referring to hypomethylation signal which persists in somatic tissues. We have edited the text to specify each epigenetic state that we refer to in these sentences on page 6:

“Taken together, our results suggest a route towards pluripotency that involves a small subset of pluripotency-associated genes and selected *cis*-regulatory elements that display unique **H3K27ac, H3K4me2 and DNA accessibility dynamics during differentiation** in our model, as well as a surprising epigenetic memory of the pluripotent state that persists as focal hypomethylation throughout development.”

- Line 309 ‘discrete molecular determination event’ – the authors defence of this description is not satisfactory. While they do show the basic timing of when determination/commitment has occurred, they do not really demonstrate it is a ‘discrete event’.

Considering the reviewer’s suggestion and as we do not know the exact time-point when this event occurs, we have now edited the sentence accordingly on page 7:

“The period between when pluripotency can be restored by Serum/LIF alone and when this OSKM-resistant barrier is imposed, points to a molecular determination event during early differentiation that is subsequently reflected in the low efficiency and extended latency of somatic cell reprogramming.”

- line 314. ‘appears to dictate’. There is no evidence for this claim. I would suggest ‘correlates with’. In their rebuttal the authors claim that they have shown that exogenous OCT4 is required for reprogramming. To be clear, they’ve shown that exogenous OSKM is required. In their experimental setup it is possible that SKM (or a combination of different factors) might be able to induce reprogramming – especially as Oct4 protein appears to be present for quite some time even in the -Dox condition (Supp Figure 1).

We have edited the respective sentence appropriately on page 7 (Discussion):

“OCT4’s ability to access these regions **correlates with** the reprogramming outcome and may represent the bottleneck through which re-establishment of pluripotency is initiated.”

- line 326 ‘delayed silencing during early differentiation’ – what exactly does this refer to? Please clarify and use a more specific description.

By “delayed silencing during differentiation”, we refer to ATAC-seq, H3K27ac and H3K4me2 collectively. As can be seen from **Fig. 4d** and new **Supplementary Fig. 3b**, reaccessed regions show significantly higher levels ATAC, H3K27ac and H3K4me2 as compared to exclusive set at 48h and 96h. This delay in complete silencing of these regions is what we refer to here. We have made changes to explicitly specify the same in the manuscript text on page 8:

“Reprogramming-associated OCT4 targets are distinguishable by several notable characteristics, including delayed silencing reflected **by the stronger persistent euchromatic signature**, a higher than genomic average CpG density, and a unique combination of transcription factor motifs. Ectopic binding by OCT4 at reaccessed regions is more distinguishable by the dual presence of co-factor motifs associated with both pluripotency and early differentiation, indicating that a combinatorial logic may dictate the differential accessibility of these regions during high- and low efficiency reprogramming.”

- line 331-2 – ‘distinct and persistent epigenetic state in fully differentiated adult tissues’ – this does seem to be a slight overstatement. The authors have only really shown a DNA methylation difference in adult tissues. Otherwise, it remains possible that the epigenetic state may actually be quite similar. In addition, in the Abstract a similar claim (‘epigenetic signature during... in vivo differentiation’) is made: this also seems too general a statement, particularly for inclusion in the Abstract.

We have edited the sentence to specify the same at page 8 (Discussion):

“Part of the pluripotency associated transcriptional and epigenetic program is also utilized during primordial germ cell (PGC) development⁴⁰⁻⁴³ leading us to speculate that these regions could be important for PGC development. **In this model, preserving germline potential within the pluripotent epiblast would require protection of critical regions during global genome remethylation, leading to a persistent epigenetic memory that is carried into rest of the soma.**”

- Line 342-346. I agree that the authors’ data is interesting in the context of cell state transitions. In my original review I suggested the authors put their work in the context of two recent papers from Austin Smith’s lab, which the authors give brief mention to in the introduction. In fact, the Discussion would be the ideal place to comment further on this. In particular how their ‘discrete moments of specification and determination’ might map onto the cell states described by Smith as naïve, formative (Smith, Development 2017) and primed – which would appear to be extremely relevant here.

We have now included this in the discussion at page 8:

“The transition state in our system emerges as the pluripotency network is dismantled but precedes establishment of a differentiated cellular identity. As such, it aligns principally with the recently described “formative pluripotency” stage that separates naïve pluripotency from lineage specification during gastrulation^{23,39}.”

Minor comments:

- the image quality in Figure 2c may not be suitable for publication.

We have now changed Fig. 2c with a higher quality image.

- Line 174 – 176 ‘shows limited binding to a minimal set of constitutively bound regions with overall low enrichment’ - this statement/description is difficult to understand. Please reword or add further explanation of the pattern binding pattern.

We thank the reviewer for pointing this out and now we have edited the sentence to make it simpler to read:

“Interestingly, OCT4 binding appears to be globally reorganized during the transient phase, which is reflected by **low enrichment within either the pluripotent- or differentiation-specific sets** (Fig. 4a and Supplementary Fig. 2a).”

REVIEWERS' COMMENTS:

Reviewer #2 (Remarks to the Author):

The manuscript is much improved. The findings will be of interest to the reprogramming field, and also labs working on ES cell commitment. I believe that although there are some caveats to the data, these are now clearly communicated.

My remaining comments are limited to minor textual amendments and referencing corrections:

1. Line 52: 'endogenous' – I understand what the authors are trying to communicate, but I am not convinced endogenous is the correct term here – all binding is endogenous. The sentence could be reworked, or the word simply removed.
2. Line 54: As I'm sure the authors are aware the role of Mbd3 in reprogramming is highly contentious. I would modify the claim made and balance it with an appropriate citation (such as Santos et al., 2014) or remove it.
3. Line 343: Smith, 2017 would be a more appropriate reference than 39.

References:

Smith, A. (2017). Formative pluripotency: the executive phase in a developmental continuum. *Development* 144, 365–373.

Santos, dos, R.L., Tosti, L., Radzsheuskaya, A., Caballero, I.M., Kaji, K., Hendrich, B., and Silva, J.C.R. (2014). MBD3/NuRD facilitates induction of pluripotency in a context-dependent manner. *Cell Stem Cell* 15, 102–110.

REVIEWERS' COMMENTS (grey) and last responses (black).

Reviewer #2 (Remarks to the Author):

The manuscript is much improved. The findings will be of interest to the reprogramming field, and also labs working on ES cell commitment. I believe that although there are some caveats to the data, these are now clearly communicated.

My remaining comments are limited to minor textual amendments and referencing corrections:

1. Line 52: 'endogenous' – I understand what the authors are trying to communicate, but I am not convinced endogenous is the correct term here – all binding is endogenous. The sentence could be reworked, or the word simply removed.

Answer: As requested by the reviewer, we have removed the word.

2. Line 54: As I'm sure the authors are aware the role of Mbd3 in reprogramming is highly contentious. I would modify the claim made and balance it with an appropriate citation (such as Santos et al., 2014) or remove it.

Answer: As suggested by the reviewer, we have modified the sentence as follows:

For instance, Mbd3 depletion has been shown to result in high reprogramming efficiencies by preventing counterproductive repression of OSKM targets by the NuRD complex¹³. Alternatively, a context dependent facilitatory role for Mbd3 in reprogramming has also been reported²³.

3. Line 343: Smith, 2017 would be a more appropriate reference than 39.

Answer: We have now added the reference mentioned by the reviewer.

References:

Smith, A. (2017). Formative pluripotency: the executive phase in a developmental continuum. *Development* 144, 365–373.

Santos, dos, R.L., Tosti, L., Radzsheuskaya, A., Caballero, I.M., Kaji, K., Hendrich, B., and Silva, J.C.R. (2014). MBD3/NuRD facilitates induction of pluripotency in a context-dependent manner. *Cell Stem Cell* 15, 102–110.